# Microbiota assembly, structure, and dynamics among Tsimane horticulturalists of the Bolivian Amazon

Daniel D. Sprockett[1], Melanie Martin [2,3], Elizabeth K. Costello[4], Adam R. Burns[4], Susan P. Holmes[5], Michael D. Gurven[3,6] & David A. Relman [1,4,7 ✉]

Selective and neutral forces shape human microbiota assembly in early life. The Tsimane are an indigenous Bolivian population with infant care-associated behaviors predicted to increase mother-infant microbial dispersal. Here, we characterize microbial community assembly in 47 infant-mother pairs from six Tsimane villages, using 16S rRNA gene amplicon sequencing of longitudinal stool and tongue swab samples. We find that infant consumption of dairy products, vegetables, and chicha (a fermented drink inoculated with oral microbes) is associated with stool microbiota composition. In stool and tongue samples, microbes shared between mothers and infants are more abundant than non-shared microbes. Using a neutral model of community assembly, we find that neutral processes alone explain the prevalence of 79% of infant-colonizing microbes, but explain microbial prevalence less well in adults from river villages with more regular access to markets. Our results underscore the importance of neutral forces during microbiota assembly. Changing lifestyle factors may alter traditional modes of microbiota assembly by decreasing the role of neutral processes.

[1] Department of Microbiology & Immunology, Stanford University School of Medicine, Stanford, CA 94305, USA. [2] Department of Anthropology, University of Washington, Seattle, WA 98195, USA. [3] Department of Anthropology, University of California Santa Barbara, Santa Barbara, CA 93106, USA. [4] Department of Medicine, Stanford University School of Medicine, Stanford, CA 94305, USA. [5] Department of Statistics, Stanford University, Stanford, CA 94305, USA. [6] Broom Center for Demography, University of California Santa Barbara, Santa Barbara, CA 93106, USA. [7] Infectious Diseases Section, Veterans Affairs Palo Alto Health Care System, Palo Alto, CA 94304, USA. ✉email: relman@stanford.edu

The human body becomes colonized in early life by microbial communities that are involved in several key metabolic and immunologic processes, including nutrient acquisition, immune programming, and pathogen exclusion. While these communities are relatively simple at birth, they grow in complexity until reaching an adult-like configuration within the first few years of life[1,2]. This rapid and dynamic period of community assembly is characterized by broad shifts in community structure[3] that arise from both deterministic processes like host-driven ecological selection[4], as well as neutral processes like dispersal and demographic stochasticity[5–7]. Yet, the relative contributions of deterministic and neutral processes to early life microbiota assembly—and the extent to which different environmental factors may moderate these effects—remain largely unexplored.

In order to assess how transmission dynamics might affect community assembly, we focused on the Tsimane people, an indigenous forager-horticulturalist population inhabiting the Bolivian Amazon basin[8]. The Tsimane practice several infant care-associated behaviors (ICABs) that potentially increase the likelihood that maternally derived microbes disperse to their offspring. For example, all Tsimane infants are vaginally birthed at home, and mothers carry infants in slings during the day and share their beds with their infants at night. Infants are breastfed "on demand" 24 h a day, and the period of exclusive breastfeeding lasts about 4 months[9,10], although typically, weaning is not complete until around 27 months of age[9]. During the introduction of complementary foods, mothers frequently premasticate, or pre-chew, foods such as rice, plantain, meat, or fish before depositing them into the mouths of their children. The most commonly consumed liquids are water (usually sourced from a nearby river), water mixed with sugar or fruit juice[11], and chicha, a fermented drink made from manioc, corn, or plantain. Chicha made from manioc and corn is inoculated with saliva; women chew and expectorate pieces of manioc during preparation, and serve the drink without cooking. Similar chicha preparations have been shown to contain high loads of diverse lactic acid bacteria and yeast species[12–16].

In comparison with industrialized societies, microbial exposure is high among the Tsimane. They lack access to improved water sources and have frequent exposure to parasites with high rates of early life morbidity and mortality owing to infectious disease[17–19]. Previous studies have reported that Tsimane children have high levels of C-reactive protein and other inflammatory markers, consistent with a high pathogen burden[19–22]. This confluence of high exposure to maternal and environmental sources of microbes and high immune system activation suggests high rates of microbial dispersal to Tsimane infants from family and environmental sources.

In recent years, rapid lifestyle changes have taken place in the Tsimane population due to increased access to market goods, cash economies, wage labor, education, and medicine[8]. These changes have not yet affected ICABs[10], and the Tsimane continue to subsist primarily on horticulture and foraging, remain highly active, and exhibit negligible cardiometabolic health risks[23]. However, secular trends have documented increasing sedentarism, BMI, LDL cholesterol, and consumption of sugar, vegetable oil, and processed foods—particularly for Tsimane residing closer to market towns[23–25]. Thus, there are narrowing opportunities to examine patterns and processes of host-associated microbiota assembly in the context of nonindustrialized Tsimane lifestyle factors and local ecological diversity.

Here, we report on microbiota data that we generated as part of a long-term study of Tsimane health and life history[8]. We first examined microbiota structure from paired stool samples ($n = 283$) and tongue swabs ($n = 117$) collected longitudinally from 47 Tsimane mother–infant dyads between 2012 and 2013, one of the largest sample collections of this type to date from a non-industrialized setting. We then placed the patterns of microbiota assembly that we observed into a global context by comparing Tsimane infants to previously studied infants from Bangladesh[26] and Finland[27]. Finally, we expanded our cohort by including data we generated from fecal samples collected in 2009 from 73 Tsimane individuals ranging in age from 1 to 58 years, enabling us to examine temporal and regional variation in microbiota composition that may be associated with seasonal differences, differences in local ecology, and/or greater access to regional markets. We found that even though neutral processes could largely explain patterns of infant microbiota assembly, these patterns were also consistent with an effect of ICABs. Stool bacterial taxa were associated with greater market access, and displayed altered dispersal patterns during infant colonization.

## Results

**Samples and subjects**. To explore whether ICABs are associated with patterns of microbial colonization in young children, stool samples (reflecting the distal gut) and swab samples from the dorsum of the tongue were collected from 47 Tsimane families living in six villages located along the Maniqui River in the Bolivian lowlands of the Amazon basin. Samples were collected from infants (0–2 years of age) and mothers (14 or more years of age) using a mixed longitudinal design (Supplementary Table 1, Fig. 1a).

**Infant and maternal microbiota dynamics following birth**. Overall, infant stool samples had high relative abundances of Bifidobacteriaceae, Enterobacteriaceae, and Veillonellaceae, while adult stool samples were dominated by Ruminococcaceae, Prevotellaceae, and Lachnospiraceae (Supplementary Fig. 1). On the tongue, both infants and adults had high relative abundances of Streptococcaceae, Veillonellaceae, and Micrococcaceae, although adult tongue samples also exhibited high relative abundances of Pasteurellaceae and Neisseriaceae (Supplementary Fig. 2). In addition, bacterial diversity increased with age in both stool and on the tongue of infants over the first 18 months of life (Fig. 1b, c). We used a linear mixed-effects model that accounted for the longitudinal structure of these data by treating the subject as a random effect. It demonstrated that diversity was positively correlated with age in both stool ($P = 1.08 \times 10^{-11}$, conditional $R^2 = 0.481$) and tongue communities ($P = .013$, conditional $R^2 = 0.397$), although diversity increased at a faster rate in the stool samples (Fig. 1b, c; ANOVA, $P = .004$). The diversity of the maternal stool and dorsal tongue communities was stable during this time period (Fig. 1b, c, $P = .065$ and .099, respectively), as has been observed previously[28].

Variation in bacterial community composition among samples from mothers and infants was primarily explained by body site (13.7%, PERMANOVA, 1000 permutations, $P < .001$) and host age group (infant or adult, 10.5%, PERMANOVA, 1000 permutations, $P < .001$). Village type (proximal or distal to the regional market) was also a significant factor (PERMANOVA, 1000 permutations, $P < .01$), although it only explained 0.48% of the variation. Microbial communities colonizing the infant gut were distinct at all ages from those of the tongue and became increasingly differentiated with time (Fig. 1d), but were never as distinct as those of adults, owing largely to incomplete maturation of these infant stool microbiotas by 18 months of age. Despite their apparent immaturity, infant stool microbiotas shared significantly more bacterial taxa with those of the mother than with unrelated adults (Wilcoxon rank-sum test, $P < .05$, Supplementary Fig. 3). While the tongue swabs of 16–18 month olds harbored microbial communities similar in composition to adult tongue swabs

# ARTICLE

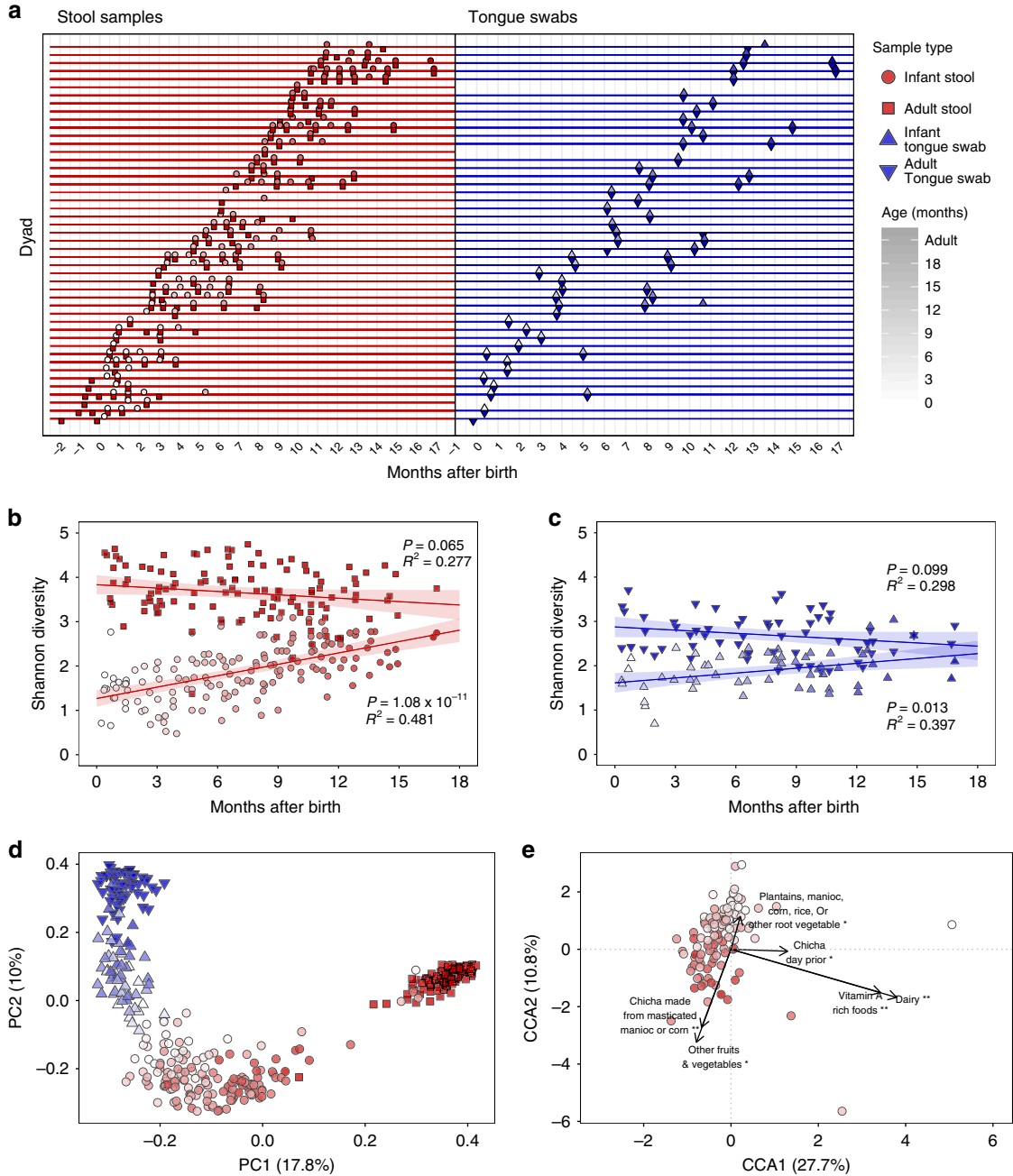

**Fig. 1 Infant microbiota dynamics and dietary factors associated with maturation. a** Timeline denoting fecal sample and tongue swab collections for each mother–child dyad relative to the infant's birth date. For maternal samples, time refers to the immediate postpartum period. Infant stool samples are circles, adult stool samples are squares, infant and adult tongue swabs are triangles that point up or down, respectively. Shapes are colored by sample type (red = stool, blue = tongue swabs), and the color darkens as the subject's age increases. Shapes and colors are consistent across **a–e**. **b** Shannon diversity index regressed against the time since the infant's birth for stool samples (adult stool: $P = .065$, $R^2 = 0.277$; infant stool: $P = 1.08 \times 10^{-11}$, $R^2 = 0.481$). Lines indicate the linear mixed-effects regression of diversity on time since delivery, while treating subject as a random effect. The shading indicates the 95% confidence interval. The conditional $R^2$ describes the proportion of variation explained by both the fixed and random factors, and was calculated using the R package, "piecewiseSEM". **c** Shannon diversity index regressed against the time since the infant's birth for tongue swab samples (adult tongue swabs: $P = .099$, $R^2 = 0.298$; infant tongue swabs: $P = .013$, $R^2 = 0.397$). Figure details are the same as in (**b**). **d** Principal coordinate analysis (PCoA) using a distance matrix calculated using the Jaccard similarity index of microbiota taxa composition in stool samples and tongue swabs from Tsimane dyads. **e** A partial canonical correspondence analysis (CCA) of ASV abundances, constrained against a matrix of diet survey data. The effects of infant age and village were controlled using a conditioning matrix. Significance was assessed using an ANOVA-like permutation test with 1000 permutations. Source data are provided in the Source Data file.

(Fig. 1d), those communities were not more similar to those of their mothers than they were to those of unrelated adults (Supplementary Fig. 3), in contrast to the stool communities. This supports the findings of a previous cross-sectional study showing

that the oral microbiotas of Tsimane infants (9–24 months old) were not more similar to those of their mothers than they were to unrelated adults, despite frequent maternal premastication[29]. The number of amplicon sequence variants (ASVs) shared between

body sites within infants was highest immediately after birth (33.3%), and then declined with time (Supplementary Fig. 4). However, the ASV overlap between body sites in 16–18 month olds was significantly higher than in adults (Supplementary Fig. 4, Wilcoxon rank-sum test, $P = .03$).

Growth faltering in the Tsimane relative to international standards might be due to pathogen burden, rather than lack of nutrition, as indicated by a much higher prevalence of height-for-age $z$-scores (HAZ) $< -2$ SD (53%), than for weight-for-age (WAZ, 16%) or weight-for-height $z$-scores (WHZ 9%) $< -2$ SD in children aged 2–5 years[30]. Within the current study cohort of 47 Tsimane children, only one subject consistently exhibited HAZ and WAZ scores below $-2$ SD. Five other subjects had HAZ or WAZ scores that crossed from greater than to less than $-2$ SD after one year of age. All infants in the longitudinal cohort were breastfeeding at all study time points. However, among non-exclusively breastfed infants over 6 months of age, 63.8% of samples were collected at times when the infant had a minimum dietary diversity (MDD) score of $< 4$ (Supplementary Fig. 5), indicating potential micronutrient inadequacy in their complementary foods[31] (see "Methods" for more details). A single linear mixed-effects model using MDD and age as predictors showed that infants' MDD scores were positively correlated with infant stool microbiota diversity (Supplementary Fig. 5, $P < .001$, conditional $R^2 = 0.374$), and accounted for a larger effect than age ($\beta_{MDD} = 1.2$ vs. $\beta_{Age\ (months)} = 0.6$, $P < .05$ for both) on infant stool diversity. Chicha consumption, as well as vegetable and dairy consumption, was significantly associated with stool microbiota composition, even after controlling for age and village (Fig. 1e). Interestingly, consuming chicha made from manioc or corn had an opposite association with the infant stool microbiota than did consuming manioc or corn itself. Fecal neopterin, a marker of interferon-activated monocytes and macrophages, and of intestinal inflammation[32], was associated with significantly lower infant stool diversity, even when controlling for age (Supplementary Fig. 5, $P = .002$, conditional $R^2 = 0.333$).

**Potential sources of infant-colonizing microbes**. To identify potential sources of infant-colonizing microbes, we first quantified the overlap between infant and adult microbiotas. 84.5% of the ASVs found in all infant stool samples at all time points were also found in at least one adult stool sample, while 78.3% of infant oral ASVs were also found in at least one adult oral sample. On average, 46.2% of the microbes colonizing an infant's stool or tongue were also found in their mother's samples from the same body site. This percentage increased in stool samples as the infant became older (Supplementary Fig. 6, blue line plus green line in panel a), increasing from 38.4% in 0–6 month olds to 50.5% in 12–18 month olds (Wilcoxon rank-sum test, $P = .0013$); in tongue swabs, the increase from 59.6% in 0–6 month olds to 63.8% in 12–18 month olds was not statistically significant (Wilcoxon rank-sum test, $P = .52$). However, the vast majority of these shared microbes were also found in other adults living in the same village as the infant (Supplementary Fig. 6, green line in panel a), reflecting a high degree of microbial sharing among nearby adults, similar to what has been observed in other indigenous populations[33]. Mothers from the same village shared significantly more stool microbes, but not more tongue microbes, than did mothers from different villages (Supplementary Fig. 6, Wilcoxon rank-sum test, $P < 0.001$), suggesting that village ecology might play a role in structuring the stool microbiota. About half of the microbes found in an infant's stool were found in at least one of their mother's stool samples (Supplementary Fig. 6, purple line plus green line in panel c), and about half were not found in any of their mother's samples (orange line), with very

few infant stool microbes ever observed on their mother's tongue (yellow line plus green line). The same was true for infant tongue microbes, of which the majority of infant tongue ASVs were observed on their mother's tongue (Supplementary Fig. 6, yellow line plus green line in panel c) and very few were observed in their mother's stool (purple line plus green line).

Even though they accounted for only about half of the taxa in the infant stool and on the tongue, the bacteria shared within a mother–infant dyad were found at higher relative abundances in both the infants and their mothers than non-shared taxa. This difference in average relative abundance was statistically significant in infants until they reached 12 months of age, after which the relative abundances of shared and non-shared taxa did not significantly differ (Fig. 2a, c). In mothers, shared taxa remained at higher abundance during the first 18 months of the infant's life (Fig. 2a, c). At both body sites, the frequency at which a given microbe was shared within a dyad was largely a function of its relative abundance in the mother. Microbes with an average maternal abundance >0.1% were shared within a dyad at the same rate as its prevalence, while microbes at 0.01% or lower abundance in mothers were almost never shared within a dyad, even if the ASVs were highly prevalent in infants (Fig. 2b, d). Overall, microbes that were highly abundant in Tsimane mothers were more likely to be shared with their infants.

**Microbiota assembly rules**. Since many Tsimane ICABs may increase the opportunities for neutral dispersal of microbes from mothers to their children, we sought to quantify the relative contributions of deterministic and stochastic (i.e., neutral) processes during microbiota assembly. For this purpose, we used the neutral community model (NCM)[34], which has previously been applied to the assembly of many host-associated microbial communities[6,7]. The NCM predicts the prevalence of each microbe given its average relative abundance in a regional species pool (RSP). Microbes that fit the prediction are inferred to have assembled neutrally from the RSP, while microbes at higher or lower prevalence are inferred to have been under local positive or negative ecological selection, respectively. However, species whose distributions fit the neutral model may still have experienced selection, if the selection were of similar direction and magnitude to the selection present in the RSP. In addition, the NCM assumes that all microbes in a RSP have an equal per capita ability to disperse to a local area, and once established, to have equal per capita fitness, growth, and death rates[34].

We defined the RSP as all of the microbes able to disperse to and colonize a local area, which in this case was the infant stool or tongue. In order to estimate the composition of this pool, we summed all of the local communities observed at a given body site in all of the infants in the cohort, and then assessed the degree to which the observations fit the model predictions by calculating the model's root mean squared error (RMSE). RMSE is the square root of the mean of the squared differences between observed and predicted values, with 0 indicating that the data perfectly fit the model and higher values indicating a greater divergence from the model predictions. To account for the longitudinal sampling scheme, we randomly selected one sample per infant with 1000 permutations to calculate a bootstrapped estimate of model fit. Since the NCM does not account for deterministic factors and makes many simplifying assumptions regarding the functional equivalence of species, it should not be interpreted as a complete description of community assembly. Nevertheless, its ability to accurately predict a species' prevalence suggests that neutral dispersal is an important force shaping microbial community structure. The NCM also serves as a useful mechanistic null model to help identify factors leading to a divergence from

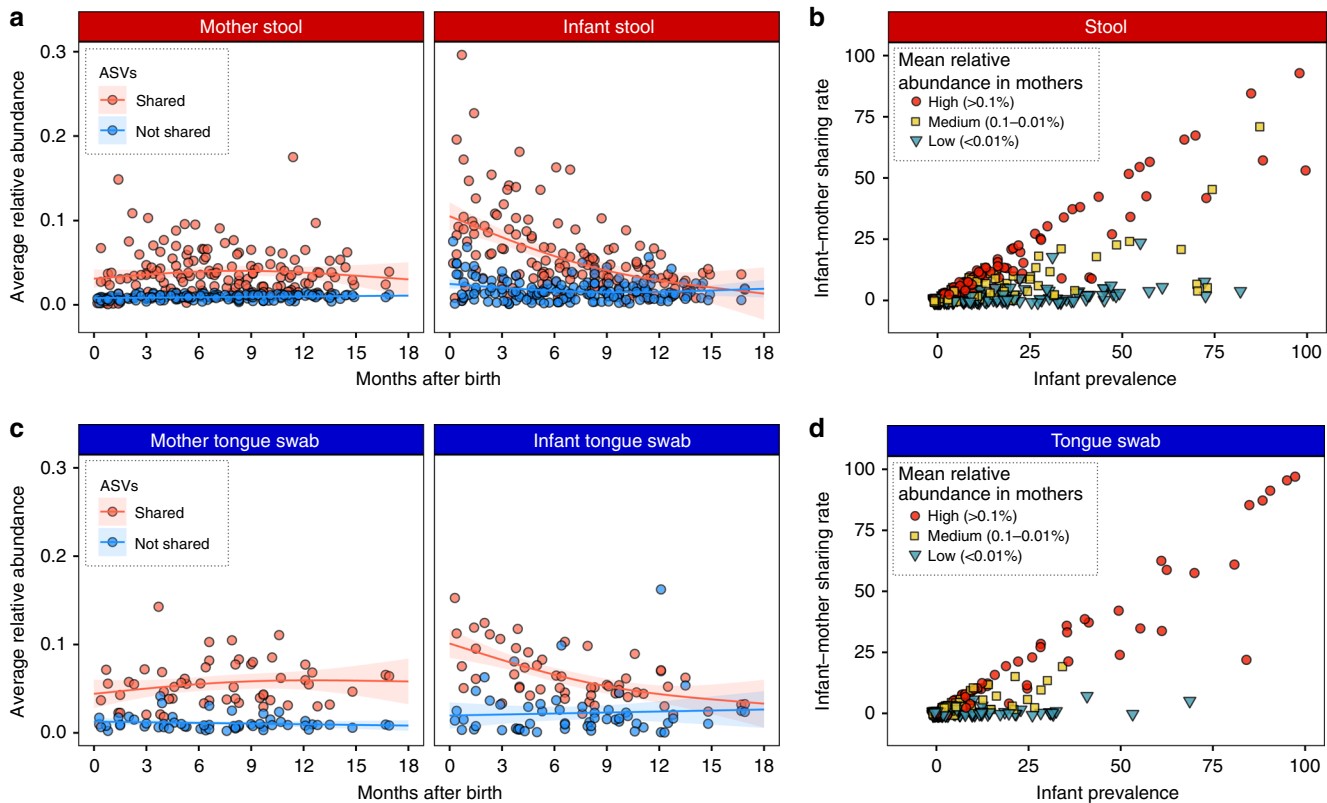

**Fig. 2 Shared microbes are highly abundant within mother–infant dyads. a** Average relative abundance of ASVs that were shared or not shared within the stool samples of an infant–mother dyad plotted against the time since infant birth. Lines represent the best fit of a generalized additive model to either shared or not shared ASVs in the mother's (left) or infant's (right) stool samples. The shading indicates the 95% confidence interval. **b** Relationship between ASV prevalence in infant gut samples and the rate at which the ASV was shared within infant–mother dyads. The points are colored according to their average relative abundance in maternal gut samples. **c** Average relative abundance of ASVs that were shared or not shared within the tongue swabs of an infant–mother dyad plotted against the time since infant birth. The shading indicates the 95% confidence interval. Figure details are the same as in (**a**). **d** Relationship between ASV prevalence in infant tongue swabs and the rate at which the ASV was shared within infant–mother dyads. Figure details are the same as in (**b**). Source data are provided in the Source Data file.

neutral dynamics or specific taxa that assemble in a nonneutral manner, such as strong competitors or active dispersers.

Of the 780 ASVs observed in infant stool, 659 were also found in at least one adult stool sample (Fig. 3a). The 484 ASVs observed in adult but not infant stool were inferred as being unable to colonize the infant gastrointestinal tract, either due to dispersal limitation or to negative selection. Microbes in infant stool fit the NCM with an average RMSE of 0.09 ± 0.006 (Figs. 3b, 1000 permutations, ±standard deviation). In total, 79.2% of infant stool ASVs were neutrally distributed in at least 90% of permutations, while only 4.0% were consistently under positive selection and 0.6% under negative selection. These low-prevalence or high-abundance taxa included obligate anaerobes such as *Prevotella, Megasphaera*, and *Bifidobacterium*, suggesting that oxygen exposure could be an impediment to transmission (Fig. 3c). The remaining 16.2% of bacterial taxa were variable in their predicted fit to the NCM across permutations. As a point of comparison, the NCM was also applied to adult stool samples. The RMSE of infant stool microbiotas was significantly lower than adult stool microbiotas (RMSE = 0.14 ± 0.003, P < 0.001), indicating that neutral processes play a more important role in shaping microbial community assembly in early life. Only 3 of the 31 consistently positively selected microbes were observed exclusively in infant stool (Fig. 3c), and of the remaining 28 ASVs, 21 were found at higher relative abundance in infant than in adult stools.

Similar to the communities found in stool, 166 of the 212 ASVs observed in infant tongue swabs were also found in at least one adult tongue swab (Supplementary Fig. 7). The 141 ASVs observed in adult but not infant tongue swabs were likely unable to colonize the infant tongue due to negative selection, especially since Tsimane ICABs may be facilitating dispersal. Microbes on the infant tongue fit the NCM with an average RMSE of 0.1 ± 0.003 (Supplementary Fig. 7, 1000 permutations, ±standard deviation). In total, 69.3% of infant tongue swab ASVs were neutrally distributed in at least 90% of permutations, while 10.8% were consistently under positive selection and 3.3% under negative selection. The remaining 16.6% of bacterial taxa were variable in their predicted fit to the NCM across permutations. Again, the NCM was also applied to adult tongue swab samples as a reference, and the RMSE of infant tongue swab microbiotas was significantly lower than adult tongue swab microbiotas (RMSE = 0.145 ± 0.004, P < 0.001), indicating that neutral processes play an important role in shaping microbial community assembly in early life in the oral cavity as well. Of the 23 consistently positively selected microbes, only 5 were observed exclusively in infant tongue swabs (Supplementary Fig. 7).

**Infant microbiota assembly in a global context**. We next sought to determine whether the assembly patterns that we observed in the Tsimane were conserved in infants from diverse cultures. We

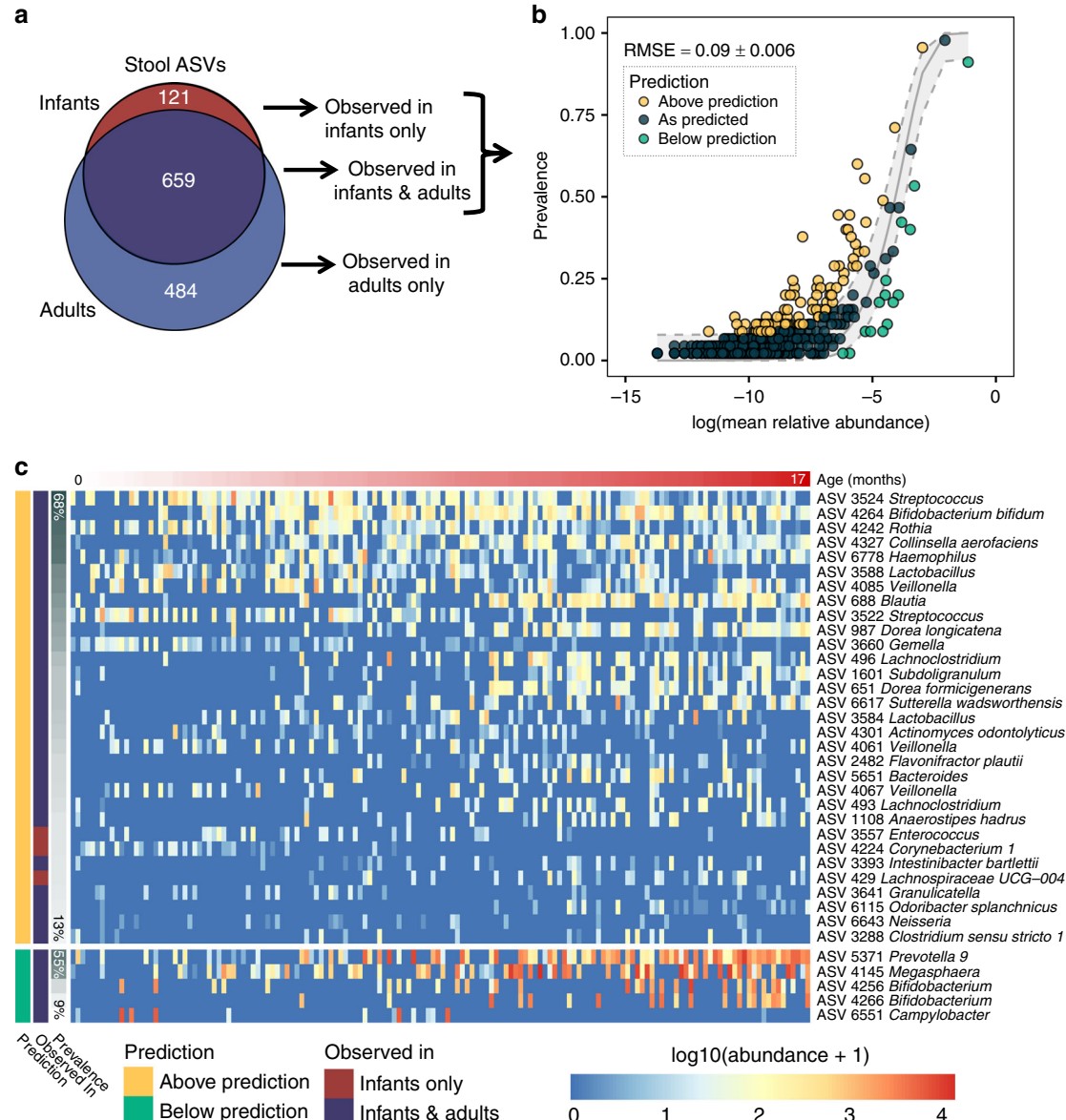

**Fig. 3 Infant stool microbiotas assemble according to neutral rules. a** Venn diagram showing the number of ASVs observed in infant and adult stool samples. Only taxa found in infant stool samples were included in panels (**b**) and (**c**). **b** Neutral community model (NCM) fit to each ASV observed in the stool of Tsimane infants. Points are colored according to whether the taxon prevalence in infant samples was above (yellow), at (blue), or below (green) the predicted prevalence according to the NCM. Average RMSE (±standard deviation) was calculated from 1000 bootstrap resamplings. **c** Heatmap of the log₁₀ normalized abundances of the ASVs from infant stool samples that were observed either to be consistently above (yellow) or below (green) their predicted prevalence in part (**b**). Rows are sorted by taxa prevalence; columns are sorted by the subject's age (months). Source data are provided in the Source Data file.

integrated our data with two publicly available datasets generated with comparable molecular methods for profiling the stool microbiota. The first of these datasets was created in a study of undernourished infants in Bangladesh[26] (we used the well-nourished control group only), and the second in a large observational study of type 1 diabetes in eastern Europe[27] (we used the Finnish infants only). The age ranges for each infant cohort were 0.2–19 months for Bolivia (Tsimane), 0–24 months for Bangladesh, and 1.2–37 months for Finland. Despite drastically different social and physical environments, access to health care, and diets, the stool microbiotas of the infants in these three studies were similar in composition at the earliest ages based on a range of distance metrics, indicating similar initial microbial colonists in societies across the globe (Fig. 4a, Supplementary Fig. 8). Based

on the Jaccard similarity index and the Bray–Curtis dissimilarity, the microbiota trajectories diverged as the children aged, becoming more distinct as they approached a population-specific adult-like state (Fig. 4a, Supplementary Fig. 8). However, both weighted and unweighted Unifrac, which are phylogenetically informed and tend to emphasize similarities at higher taxonomic levels, revealed that microbiota developmental trajectories are much more consistent between populations (Supplementary Fig. 8).

In order to determine if neutral processes operate in a similar manner among infants in different cultural settings and geographies, we divided each dataset into three-month age intervals up to 1 year of age, and then reapplied the NCM model using only the samples from that interval as the estimate of the RSP, building

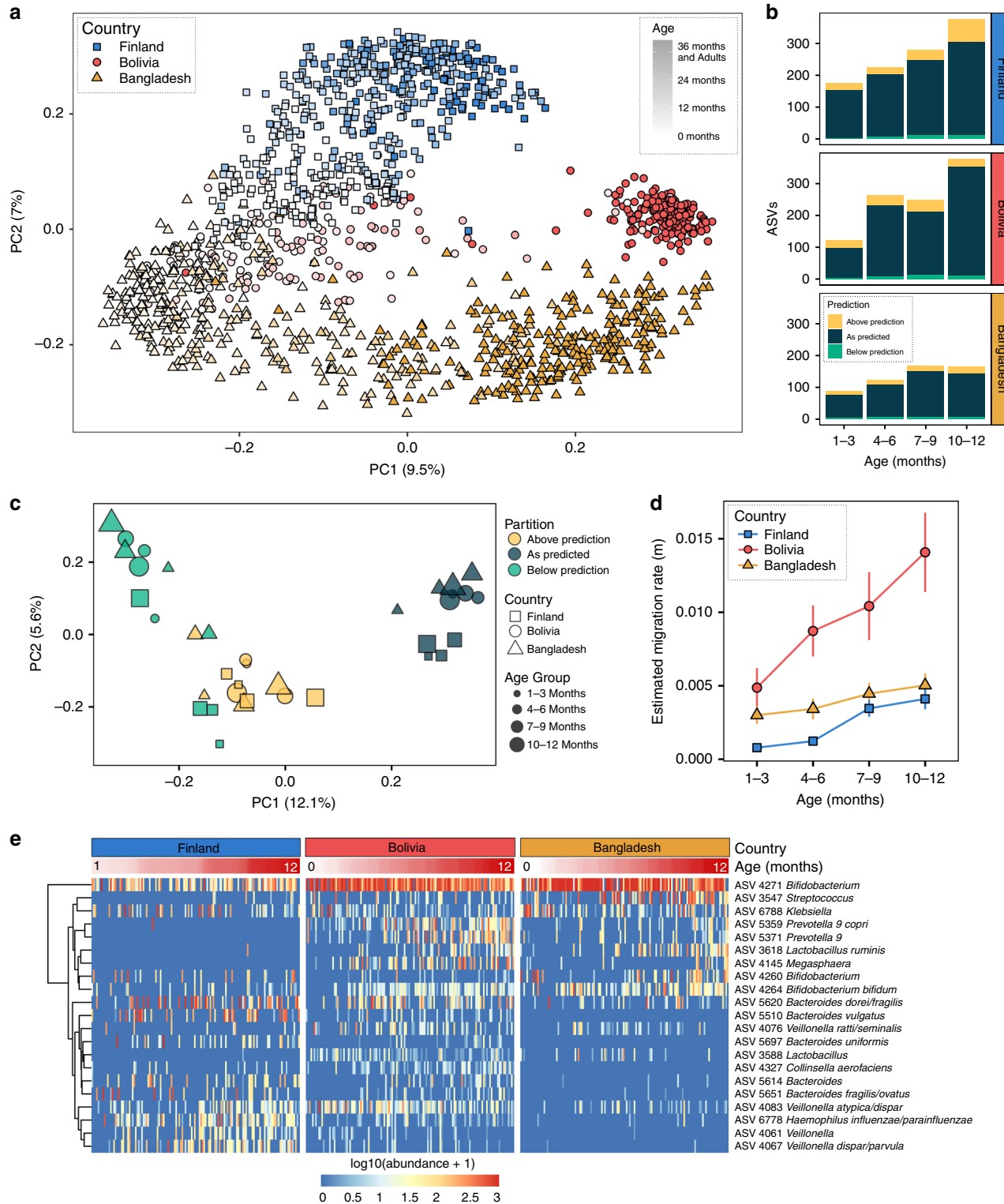

a total of 12 additional models (Supplementary Fig. 9). These datasets exhibited striking consistency in the role of neutral and nonneutral processes across both age and geography, with the majority of microbes exhibiting patterns of neutral assembly across individuals (Fig. 4b). Furthermore, by partitioning ASVs according to country, age group, and how well they fit the neutral model, we found similar sets of neutral and nonneutral microbes across geography and ontogeny (Fig. 4c, Jaccard distance/PCoA). A heatmap displaying only positively selected microbes revealed

*Bifidobacterium* (ASV 4271) as highly prevalent and abundant in all three cohorts (Fig. 4e). Country-specific patterns were also apparent, including four *Bacteroides* ASVs (5510, 5697, 5614, and 5651), *Collinsella aerofaciens* (4327), one *Haemophilus* ASV (6778), and two *Veillonella* ASVs (4061 and 4067) that were prevalent in Finnish and Bolivian (Tsimane), but not Bangladeshi infants, as well as two *Prevotella* ASVs (5359 and 5371), *Lactobacillus ruminis* (3618), and one *Megasphaera* ASV (4145) that increased in prevalence with age in both the Bolivian

**Fig. 4 Patterns of selection and neutral assembly are consistent across societies. a** Principal coordinate analysis (PCoA) using the Jaccard similarity index of stool samples from Bolivian (Tsimane) dyads, as well as previously published 16S rRNA datasets of Bangladeshi and Finnish subjects. Points are shaped and colored according to country, and the color darkens with increased age of the subject (Finland—blue squares, Bolivia (Tsimane)—red circles, Bangladesh—yellow triangles). See Supplementary Fig. 8 for additional ordinations. **b** A summary of the proportion of ASVs present in each country and age group that fit the model (blue), or was above (yellow) or below (green) the model's prediction. See Supplementary Fig. 9 for details. **c** Principal coordinate analysis using the Jaccard distance for each of the three data partitions of four age groups from three countries (36 measurements in total). The color indicates the predicted fit, the shape indicates the country whence the samples were collected, and the size of each shape is scaled to indicate the age group of the subject. **d** The estimated migration rate (m), or the probability that a random loss of an individual in a local community will be replaced by dispersal from the metacommunity, for each country and age group. This parameter was fit for each country and age group using nonlinear least-squares fitting. Vertical lines represent 95% confidence intervals around the model predictions using the Wilson score interval. Colors and shapes are the same as in (**a**). n samples per country for each age group: 1–3 months, Bolivia 32, Bangladesh 56, Finland 65; 4–6 months, Bolivia 37, Bangladesh 61, Finland 55; 7–9 months, Bolivia 33, Bangladesh 56, Finland 41; 10–12 months, Bolivia 30, Bangladesh 53, Finland 51. **e** A heatmap of the $\log_{10}$ normalized abundances of ASVs that were estimated to be above their predicted prevalence given their average relative abundance in all 12 yellow partitions in (**c**). Infant cohorts were all randomly subsampled to the same number of samples for clarity. Rows were clustered on the y-axis using Ward's minimum variance method. Columns were sorted by infant age (0–12 months). Source data are provided in the Source Data file.

(Tsimane) and Bangladeshi infants, while remaining nearly absent in Finnish infants (Fig. 4e). Interestingly, *Bacteroides* and *Prevotella* have been identified as biomarkers for Western and non-Western lifestyles[35], respectively, yet both were prevalent in Tsimane infants.

The NCM also predicts the probability that when an individual microbe is removed from the local community, it is replaced by a microbe from the RSP rather than by reproduction from within the local community[7,34]. This migration term, m, increased with age in all three populations, although it increased significantly more in the Tsimane infants (Fig. 4d). This significantly higher rate of microbial migration into the Tsimane infant gastro-intestinal tract is consistent with their high exposure to maternal and environmental microbes.

**Local ecology, market access, and Tsimane microbiota structure.** In recent decades, the Tsimane have experienced profound changes to their culture and life history, including increased access to processed foods in markets and greater access to modern medical care. Shorter distance and ease of transport to local market towns is associated with lower infant mortality[18], but also higher intake of processed food[24], increased obesity and metabolic syndrome[36], and higher levels of urinary phthalates[37], suggesting that increased market access can have important health impacts. In order to understand how these lifestyle shifts may be associated with the assembly of the gastrointestinal microbiota, we compared the stool microbiota structure of adults living in nine Tsimane villages. These villages belonged to two village ecotypes, 'river' and 'forest', which affect some seasonal subsistence activities (e.g., fishing) and frequency of access to market (Supplementary Table 1, Fig. 5a). During the sample collection period in river villages (September 2012–March 2013), proximal river villages (≤20 km to market) could regularly access the market via 1–3 h boat rides combined with approximately 1 h taxi trips during the dry season (approximately June–August). Distal river villages (>20 km to market) had access to market via 6–12 h trips on foot or by boat. Inhabitants of forest villages (>40 km from market) can only access market towns via 2–3 day trips on foot, or hitchhiking on infrequently passing vehicles (approximately 6 h). Samples from forest villages were collected during the dry season (July 2009), the only time roads are accessible (Supplementary Table 1).

A principal coordinates analysis on the weighted Unifrac distances among these microbiota data showed significant clustering of stool samples by ecotype of the village in which the subject lived (Supplementary Fig. 10, PERMANOVA, P = .01). Additional PCoAs based on unweighted Unifrac distance, Jaccard similarity index, and Bray–Curtis dissimilarity all showed

similar results (Supplementary Fig. 10, PERMANOVA, P < .001 for all). PC1, which explained 25.5% of the variation in the data, was significantly correlated with the diversity of the microbiotas (Fig. 5b, P = .005, $R^2 = 0.277$). To evaluate whether these relationships might reflect an effect of season, local ecology, and/or more regular market access on assembly processes, we applied the NCM to stool samples from adults living in the two different village ecotypes with the regional species pool estimated from all of the samples from that age group and village ecotype, and found that the goodness of fit is lower in samples from river villages (t test, P < .003). This suggests that differences in local ecology and/or more regular market access, perhaps through increased consumption of processed foods (e.g., breads, pasta, and refined sugar) and/or medicine (e.g., antibiotics) may increase the role of ecological selection and diminish the role of stochastic dispersal during assembly of the gastrointestinal microbiota.

To identify how even subtle ecological and economic differences among villagers might be associated with stool microbiota structure, we used a phylogeny-based implementation of linear discriminant analysis. This approach uses both the leaves of the phylogenetic tree (i.e., the ASVs), as well as the nodes, to identify linear combinations of features that characterize two or more sample classes. The first model, built to discriminate between proximal and distal river villages, selected three predictors corresponding to ten ASVs. Nine of those ASVs, including two in the family Muribaculaceae (ASVs 5801 and 5805) and seven in the family Bacteroidaceae (ASVs 5614, 5639, 5697, 5510, 5620, 5651, and 5716) were discriminatory for the distal river samples, while only one ASV, in the family Prevotellaceae (ASV 5418), was discriminatory for the proximal river samples (Fig. 5c). The small number of proximal village-discriminatory features is consistent with studies suggesting that life in a high resource setting often leads to loss of bacterial diversity in the gastrointestinal tract[38,39]. Despite the small number of optimum features following 10-fold cross validation, the model accurately predicted village type for 87.3% of samples (Supplementary Fig. 11).

The second model, built to discriminate between river and forest villages, identified 24 predictors corresponding to 56 ASVs, and correctly identified the village type for 91.8% of the samples (Supplementary Fig. 11). River villages were associated with ASVs in the bacterial families Prevotellaceae (18 ASVs), Lachnospiraceae (13 ASVs), Veillonellaceae (4 ASVs), Acidaminococcaceae (2 ASVs), and Ruminococcaceae (2 ASVs). Indicator taxa for Tsimane forest villages included ASVs in the bacterial families Ruminococcaceae (4 ASVs), Clostridiales Family XIII (2 ASVs), Lachnospiraceae (1 ASV), Muribaculaceae (1 ASV), Peptostrep-tococcaceae (1 ASV), and Christensenellaceae (7 ASVs), as well as

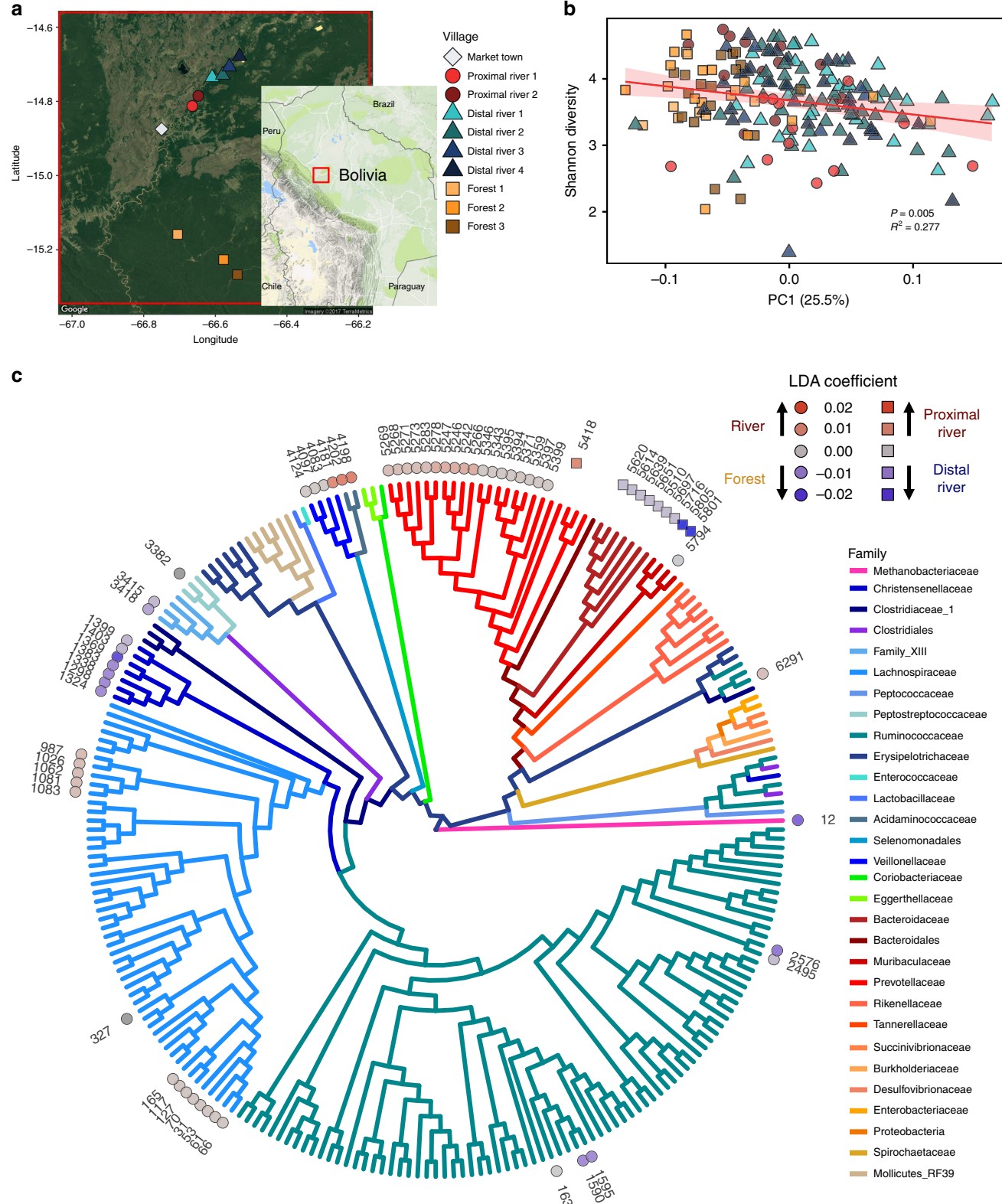

1 archaeal ASV in the family Methanobacteriaceae (1 ASV) (Fig. 5c).

Christensenellaceae and Methanobacteriaceae were previously identified in a study of adult twins as members of a co-occurrence network of heritable stool microbes that were also associated with body leanness[40,41]. Since Tsimane have low but increasing rates of obesity[23], we sought to determine if a similar network were present in the Tsimane adults. The optimal network topology constructed from adult stool microbial communities contained a subnetwork involving the most highly connected node, ASV 12 (*Methanobrevibacter*), as well as five ASVs in family Christensenellaceae (Fig. 6a), indicating strong associations among these taxa that are driven by $H_2$ syntrophy[42]. However, we did not observe an association between the relative abundance of the taxa

**Fig. 5 Market access is associated with Tsimane gut microbiota structure and diversity. a** A map showing the location of the nine Tsimane villages whence stool samples were collected. The map insert denotes the region of Bolivia being displayed. Map data: Google, TerraMetrics. **b** Shannon diversity index plotted against the PC1 value from Supplementary Fig. 10 ($P = .005$, $R^2 = 0.277$). The red line indicates the linear mixed-effects regression while treating subject as a random effect, and shading indicates the 95% confidence interval. The conditional $R^2$ describes the proportion of the variation explained by both fixed and random factors. Samples are colored by village and their shapes denote the village type (proximal river villages—circles, distal river villages—triangles, forest villages—squares). **c** Phylogenetic tree with ASVs found at least 10 times in at least 20% of Tsimane adult stool samples. Discriminatory ASVs were identified using the tree-based LDA algorithm in the treeDA R package. ASVs are denoted by shapes based on two sets of comparisons (circles for river vs. forest, and squares for proximal river vs. distal river), and are colored based on their discriminatory strength. ASVs are colored by the taxonomic family and significantly discriminatory ASVs are labeled with their ASV number. Source data are provided in the Source Data file.

in this subnetwork and adult BMI. This subnetwork also included the second most highly connected node, *Prevotella copri* (ASV 5359). Although these two taxa frequently co-occurred, their abundances were inversely related (Fig. 6b, $P = 1.04 \times 10^{-10}$, marginal $R^2 = 0.205$). *Methanobrevibacter* and *Prevotella* have been shown to be positively correlated in the gastrointestinal tract of humans[43] and western lowland gorillas[44], especially in the setting of seasonal, carbohydrate-rich diets. These taxa also had distinct colonization dynamics in the gastrointestinal tract of infants from Bolivia (Tsimane), Bangladesh, and Finland. *Prevotella copri*, the same ASV that was abundant in Bolivian and Bangladeshi infants but essentially absent in Finnish infants, increased in abundance beginning at approximately 6 months of age, and remained at high levels throughout adulthood (Fig. 6c). *Methanobrevibacter*, on the other hand, was not detected in these stool microbiotas until later in adolescence (Fig. 6c). However, since *Methanobrevibacter* was more common in forest villages, and these villages did not include many samples from young children, this pattern may also have been a result of ecological, seasonal, or societal differences between villages.

## Discussion

Here, we demonstrate that the Tsimane, an indigenous forager-horticulturalist population, display a distinct microbiota maturation trajectory among their infants. Their colonization dynamics may be the result of intensive ICABs. Strain-resolved investigations of mother–infant transmission have found that the majority of vertically transmitted microbes in the infant likely originate from the same body site in the mother, and that durable colonization by strains originating from other maternal body sites is less common, especially beyond the first week of life[45]. However, we observed evidence of maternal oral-to-infant gastrointestinal tract transmission among Tsimane participants, which could be a reflection of traditional intensive ICABs like frequent premastication, and consumption of chicha inoculated with maternal oral microbes. A study of Italian 4-month olds found that 9.5% of bacterial strains in their stool were also found on their mother's tongue[45], whereas we observed an average of 23.7% of 3–5 month old Tsimane infant stool ASVs on their mother's tongue. While metagenomic sequencing allows investigators to track the transmission of specific strains, our 16S rRNA profiling approach, paired with the inference of exact sequence variants, enabled us to perform broad surveys of assembly patterns that can serve as an "upper bound" for estimating transmission rates. This approach also allowed us to apply statistical models to quantify underlying ecological processes that drive host–microbe biology. One important caveat is our assumption that microbes that were shared between mothers and infants imply mother-to-infant transmission. Importantly, we cannot directly observe the direction of transmission, nor can we exclude the possibility that both mothers and children were colonized by microbes from other, unobserved sources.

The patterns of microbial prevalence and abundance fit the NCM less well in adults living in river villages with more regular

market access than those adults from forest villages that are more remote. This could suggest that the structure of the Tsimane microbiota may be changing as a result of the rapid socioeconomic shifts that they are experiencing, especially in ways that may decrease the contribution of neutral processes during childhood microbiota assembly and potentially alter interactions between highly prevalent microbes like *Methanobrevibacter* and *Prevotella copri*. Microbial community differences across villages may also reflect local ecological influences, including water sources (river versus lake or stream), pathogen exposures (increased during the rainy season), and dietary composition of foraged foods (e.g., greater fish consumption among river villages). Logistical constraints in our study design necessitated that river villages were sampled across rainy and dry seasons, but forest villages only during the dry season when accessible along a poorly maintained logging road. Differences related to regional market access may be stronger than what we observe, due to potential confounding by season. Additional longitudinal research within regions would be necessary to discern how seasonal dietary variation and frequency of market access interact to influence infant microbial assembly and shifts in adult composition.

The differences we observed across Tsimane regions are perhaps not surprising, given that differences in subsistence strategies have been associated with differences in the stool microbiota of indigenous populations[46], and relatively modest differences in lifestyle can lead to differences in the stool microbiota[47]. A recent study identified microbes that were differentially abundant in indigenous Bassa infants living in rural Nigeria versus infants living in nearby urban centers[48], hinting that the observed microbiota differences in adults from traditional societies may be influenced by early life factors[39]. Here, we suggest that the differences observed between infants living in traditional societies and those living in or near industrialized cities might be the result of reduced neutral dispersal of microbes that accompanies the hygienic, medical, or societal changes associated with increased market integration. Alternatively, these changes may be the result of maternal microbiotas that have a decreased ability to stably colonize infants. Our findings lend support to the hypothesis that familial microbial transmission in western societies deviates from traditional modes of transmission[49].

While there are substantial differences in bacterial taxa between adult populations from different regions and cultures, the prominent role of neutral assembly in infant guts is strikingly consistent. This is especially important in the transmission of commensal microbes in indigenous populations[50], since these relatively isolated societies experience low levels of microbial dispersal from outside their local communities, and exhibit a high degree of homogeneity in their microbiota composition[33]. Collectively, these observations of the microbiota in traditional populations have broad implications for understanding how ecological interactions operate during microbiota assembly, as well as how those patterns are altered due to broad societal changes that have arisen in concert with increased participation in market economies.

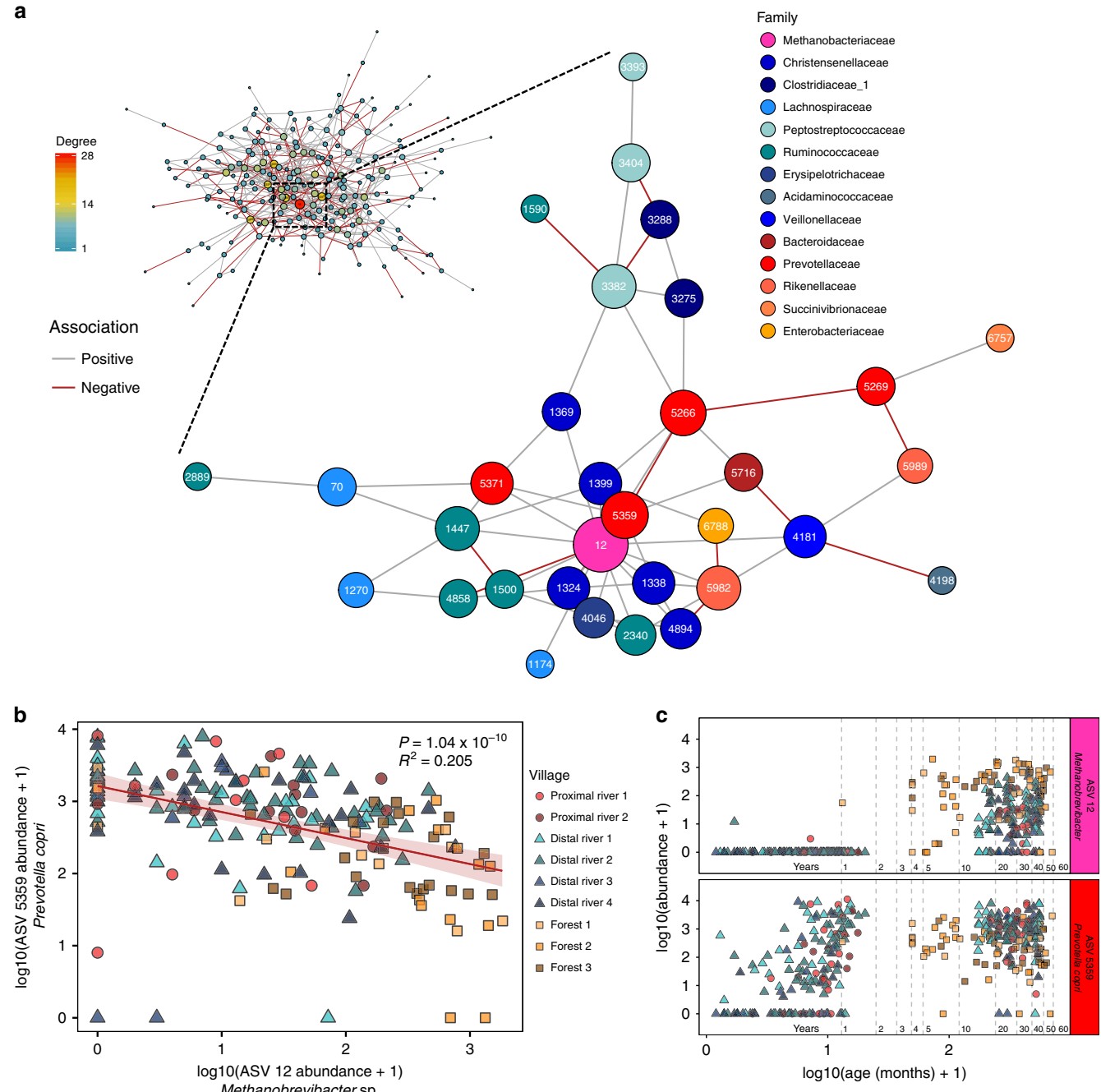

**Fig. 6 Network analysis reveals an inverse relationship between highly connected taxa. a** A network diagram of ASVs observed at least ten times in at least 20% of Tsimane adult stool samples. The size of the node was scaled to represent its degree of connectedness. The insert shows the complete network colored by network degree, while the main figure shows the highly connected subgraph colored by each ASV's taxonomic family. Positive associations between ASV nodes are colored gray, while negative associations are colored red. **b** A scatter plot of the $\log_{10}$ transformed relative abundances of the two most highly connected ASVs in the network, Methanobrevibacter spp. (ASV 12) and Prevotella copri (ASV 5359) ($P = 1.04 \times 10^{-10}$, $R^2 = 0.205$). The red line indicates the linear mixed-effects regression while treating subject as a random effect, and shading indicates the 95% confidence interval. The marginal $R^2$ describes the proportion of variation explained by the fixed factors alone, and was calculated using the R package, "piecewiseSEM". Samples are colored by village, and their shapes represent the village type (proximal river—circles, distal river—triangles, forest—squares). **c** Scatter plot of the relative abundances of Methanobrevibacter spp. (ASV 12) and Prevotella copri (ASV 5359) and the subject's age in months at the time of collection. The x-axis was $\log_{10}$ transformed for clarity when plotting both infant and adult samples. Ages 1–5, 10, 20, 30, 40, 50, and 60 years are denoted by horizontal dashed lines. Source data are provided in the Source Data file.

## Methods

**Ethics approval**. All study protocols were approved by the University of California, Santa Barbara Institutional Review Board on Human Subjects (IRB Protocols # ANTH-GU-MI-010-3U, submission ID 09-312, approved 8/21/2009; ANTH-GU-MI-010-19N, submission ID 12-354 approved 6/08/2012). Permission to conduct research was granted to the Tsimane Life History Project (THLHP) and their research affiliates. The THLHP maintains formal agreements with the local municipal government of San Borja and the Tsimane governing body. Consent was obtained from village leaders and community members during initial meetings upon starting research activities in each village. Consent was then obtained verbally from study participants prior to data collection. Mothers provided parental consent to collect samples from infants.

**Sample collection and processing**. The two sets of samples (2012–2013 and 2009) were collected as part of a long-term study of Tsimane health and life history, which also included detailed surveys of the subject's health, diet, and lifestyle. The first set of samples presented in this manuscript was collected between September 2012 and March 2013 using a mixed longitudinal design, and focused on mother–child dyads. A total of 156 stool samples and 58 tongue swabs were collected from 48 infants, and 134 stool samples and 62 tongue swabs were collected from 51 adults (see Supplementary Table 1). Samples from incomplete dyads (i.e., dyads from which samples were available only from the infant, or only from the mother) were excluded from further analysis. The remaining 47 infant–mother dyads resided in 6 villages along the Maniqui River. These complete dyads contributed 283 stool samples (infant = 153, adult = 130) and 117 tongue swabs (infant = 57, adult = 60) to the final analysis. These river villages were grouped into 'proximal' or 'distal' villages, based on their proximity to the same, nearest market town.

Samples were collected together with data on breastfeeding status, anthropometrics, and symptoms of infectious disease, as part of a study of changes in infant feeding and related maternal and infant health outcomes[9] (see Supplementary Data 1 for details). In addition, mothers were asked to recall all complementary foods consumed by their infant in the previous 24 h in order to calculate the infant's MDD score[31]. MDD is a validated population-level indicator developed by the World Health Organization to assess dietary diversity of children aged 6–23 months, and is based on the 24 h dietary recall of 7 key food groups, including grains, roots and tubers; legumes and nuts; dairy products (milk, yogurt, and cheese); flesh foods (meat, fish, poultry and liver/organ meats); eggs; vitamin-A rich fruits and vegetables; and other fruits and vegetables. The second analysis presented in this paper included samples that were collected in July 2009; a cross-sectional design specified collection of single stool samples from each of 73 Tsimane individuals ranging in age from 1 to 58 years old. These individuals resided in one of three forest villages between 31 and 68 km from the nearest regional market.

Field specimen collection protocols were devised with the logistical challenges of this population in mind, and were consistent across the 2012–2013 and 2009 cohorts. Tsimane families do not have plumbing, do not use pit toilets or diapers, and do not have access to refrigeration. Fecal samples were collected in sterile urine specimen cups. Cups were given to mothers the day before sample collection, in separate plastic bags, with specimen cup lids identified by different symbols for mother and infant. Mothers were instructed to fill the collection cups with the first bowel movement of the following day, and to keep collection cups out of the sun. Bags included sheets of paper onto which to defecate and small plastic spoons with which to handle feces, though we could not be sure that this protocol was followed. Mothers collected infant feces directly from the skin or from swaddling clothes.

Researchers returned to participants' homes between 7 and 9 a.m. to collect the specimens, making 1–2 return visits as necessary until approximately midday. Samples were transported to a field laboratory in coolers with reusable ice packs within 1–2 h of collection. Samples were homogenized in the collection cup and then partitioned into 2 ml sterile cryotubes using nonsterile wooden tongue depressors. Cryotubes were immediately stored in liquid nitrogen. Oral samples were collected from participants by research staff with a buccal cell collection swab (Catch-All™ Sample CollectionSwab; Epicentre[R]), transferred to cryotubes and stored in liquid nitrogen. All samples from both cohorts were shipped on dry ice to a laboratory in the United States, where they remained at −80 °C until they were processed.

DNA was extracted from the samples using the MOBIO PowerSoil-htp Kit (MOBIO, Carlsbad, CA) following the manufacturer's instructions, including a 2 × 10 min bead-beating step using the 0.7 mm garnet beads that were supplied by the manufacturer and the Retsch 96 Well Plate Shaker at speed 20. The V4 hypervariable region of the 16S rRNA gene was then polymerase chain reaction-amplified in triplicate using bacterial specific primers (515F: GTGCCAGCMGC CGCGGTAA and 806R: GGACTACHVGGGTWTCTAAT) that include Hamming error-correcting barcodes capable of correcting one sequencing error and detecting two sequencing errors, as well as Illumina sequencing adapters[51]. Amplicons were then pooled in equimolar ratios before being sequenced on two runs of an Illumina MiSeq 2 × 250 PE, generating a total of 35.5 million raw reads.

**16S rRNA gene sequencing and data processing**. Raw reads were denoised into ASVs using the DADA2 pipeline (dada2, version 1.14.0). Following generation of an ASV table, sequences were chimera-checked, and those remaining that were not 230–235 bps in length were removed. In addition to these data, publicly available 16S rRNA datasets from Bangladesh[26] and Finland[27] were downloaded, run through the DADA2 pipeline, and combined at the level of ASVs with the Tsimane dataset.

Taxonomic assignments were made using the RDP classifier and the SILVA nr database v132. ASVs that were not identified as Domain *Bacteria* or Domain *Archaea* were excluded from further analyses. A phylogenetic tree of the total set of ASVs was inferred using the fragment insertion function (SEPP and pplacer) in QIIME2 (version 2019.1) and the full SILVA v132 tree. An aggregated mapping file from all four of the datasets, the ASV table, the taxonomy table, and the phylogenetic tree were imported into R and combined into a single phyloseq object (phyloseq, version 1.28.0).

**Data analysis**. Both alpha-diversity (Shannon diversity index) and beta-diversity (Jaccard similarity index, Bray–Curtis dissimilarity, weighted and unweighted Unifrac) analyses were performed using the phyloseq R package. Statistical analyses were performed in R, including Wilcoxon tests performed using the "stat_compare_means" function in the R package ggpubr (version 0.2.1) and PERMANOVA tests were also performed using both the 'adonis' function in the vegan R package (version 2.5.5) and the "PermanovaG" function in the UniFrac R package (version 1.1). PCoA and canonical correspondence analysis (CCA) ordinations, boxplots, and scatter plots, were generated using the ggplot2 package in R (version 3.2.0). Heatmaps were generated using the pheatmap R package (version 1.0.12). ASV abundances were centered and log ratio-transformed before performing a PCoA on the Jaccard pairwise distances. The partial CCA was performed on centered and log transformed ASV abundances constrained against a matrix of diet survey data, including the seven MDD food groups, as well as information on meal frequency, frequency of consuming premasticated foods and liquids, frequency of consuming chicha, as well as chicha type. The effects of infant age and village were controlled using a conditioning matrix. Significance was assessed using an ANOVA-like permutation test with 999 permutations.

For the treeDA analysis in Fig. 5, ASVs were removed if they were not observed at least ten times in at least 20% of Tsimane adult stool samples. ASVs were inverse hyperbolic sine-transformed before running the tree-based LDA algorithm in the treeDA R package (version 0.0.3). The LDA score of each adult stool sample was calculated from a phylogeny-based form of linear discriminant analysis optimized with tenfold cross validation.

Network analysis was performed using the "SPIEC-EASI" function in the SpiecEasi R package (version 1.0.6) to infer an underlying graphical network model using both sparse neighborhood and inverse covariance selection[52]. ASVs were removed that were not observed at least ten times in at least 20% of Tsimane adult stool samples. The network was constructed with the SPIEC-EASI network inference algorithm using the Meinshausen–Buhlmann Neighborhood Selection method. The sizes of the nodes were scaled to represent their degree of connectedness. The distance between the nodes was determined using the force-directed layout algorithm by Fruchterman and Reingold as implemented in the "layout.fruchterman.reingold" function, while community substructure was determined using the multilevel modularity optimization algorithm for finding community structure as implemented in the "cluster_louvain" function, both available in the igraph R package (version 1.2.4.1).

**Modeling neutral community assembly**. As applied by Sloan et al.[34], neutral assembly theory assumes that all microbes in a regional species pool have an equal ability to disperse to a local area, and once established, all have equal fitness, growth, and death rates. These assumptions can be tested using a nonlinear least-squares model available in the minpack.lm R package (version 1.2-1) to predict the prevalence of a microbe in a local community based on its average relative abundance in the regional species pool. Microbial distributions that are consistent with the model's predictions are well explained by neutral dispersal, while taxa that deviate from the model likely experience either positive or negative selection. Our approach was a modified version of one used in Burns et al.[7]; modeling and plotting functions are available in the tyRa R package (version 0.1.0) available at https://danielsprockett.github.io/tyRa/.

**Reporting summary**. Further information on research design is available in the Nature Research Reporting Summary linked to this article.

## Data availability

The dataset generated as part of this current study is available at NCBI Sequence Read Archive (BioProject ID PRJNA574920 [https://www.ncbi.nlm.nih.gov/bioproject/PRJNA574920]). Additional datasets analyzed in this study are available at European Nucleotide Archive (Accession code PRJEB5482 [https://www.ebi.ac.uk/ena/data/view/PRJEB5482])[26] and NCBI Sequence Read Archive (BioProject ID PRJNA290380 [https://www.ncbi.nlm.nih.gov/bioproject/PRJNA290380/])[27]. The SILVA nr database v132 is available on the DADA2 github repository (https://benjjneb.github.io/dada2) and at www.arb-silva.de. The source data underlying all figures are provided in the "Source Data" folder, which contains two data text files, one for sample-associated data and the other for diet questionnaire data, and three R data files, two of which are used to create Fig. 5a. The third R data file contains the phyloseq object that is used to create all other figures.

## Code availability

A detailed description of these analyses, along with all datasets and analysis code, is available at the Stanford Digital Repository (https://purl.stanford.edu/tv993xn7633) and in the "Source Data" folder, which contains six R markdown files—one for each of the six figures.

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

## Acknowledgements

We would like to thank the Tsimane host villages and families that participated in this study, as well as the THLHP staff and researchers who provided invaluable assistance during field data collection. We would also like to thank David Sela at the University of Massachusetts Amherst for help with initial study design, and Alvaro Hernandez at the University of Illinois Roy J. Carver Biotechnology Center for outstanding DNA sequencing services. This research was supported by the National Science Foundation NSF BCS 0422690 (M.G.), DDIG 1232370 (M.M.), GRF DGE-114747 (D.S.), National Institute of Health grants NIH/NIGMS T32GM007276 (D.S.), NIH/NIA R01AG024119 (M.G.), NIH/NICHD K99HD074743 (E.K.C.), the Wenner-Gren Foundation (M.M.), the Thomas C. and Joan M. Merigan Endowment at Stanford University (D.A.R.), and the Chan Zuckerburg Biohub Microbiome Initiative (D.A.R.).

## Author contributions

The study was designed by E.K.C., D.A.R., M.G. and M.M.; M.M. collected all the biological samples; D.D.S. and E.K.C. processed the samples and sequenced the 16S rRNA gene amplicons. The data analysis was led by D.D.S., with assistance from M.M., A.B., E.K.C., S.P.H. and D.A.R. D.D.S. wrote the first draft of the paper, and D.D.S., M.M., A.B., E.K.C., S.P.H., M.G., and D.A.R. contributed to the final draft. All authors approved of the paper.

## Competing interests

The authors declare no competing interests.
