## [Peer Review File · Nature Communications]

Reviewers' comments:

Reviewer #1 (Remarks to the Author):

The study explores the microbiota of infants and their mothers from an indigenous Bolivian population, with a particular emphasis on the contribution of neutral processes to the assembly of the microbiota. I find the study generally interesting, in particular because it includes both a temporal and spatial view on the changes in the microbiota of the sampled individuals.

I have however two major questions regarding the application of the neutral model, which seem central to their interpretations, and which I would like to see addressed before publication.

1) What's the exact interpretation of the R^2 values given throughout the paper? As generalized R^2 values for non-linear fits do not allow for a direct interpretation as "variance explained" and are not bounded between 0 and 1, I'm wondering what a value of e.g. $R^2 = 0.38$ actually "means", other than the model fits the data "moderately well" (l. 214)? This is in particular an issue when comparing goodness of fit across different datasets and drawing central conclusions like "market integration may increase the role of selection" (l. 312) because R^2 "decreased in samples with higher levels of market access" (l. 311 and Fig 5c). Looking at the fits and R^2 I can only see that the model fits "moderately well" in all cases, but a more quantitative interpretation of the R^2 values seems difficult. Or in other words, does $R^2=0.45$ (proximal river) really indicate a significantly "less well" fit than 0.61 (forest), given that these values do not correspond to actual variance explained?

2) I find the approach of using different regional species pools (infant and mother pools) very interesting. But I have some problems with the conceptual interpretation of this.

Imagine for example that the mother pool is largely identical to the infant pool in terms of relative abundances, but contains a few additional species which are not present in infants. These species are dropped from the analysis (if I read the accompanying R code correctly?), which makes sense I guess. But because otherwise the pools are almost identical, I would expect the fit to the species actually present in infants to not change very much. By the authors logic, if I understood it correctly, in this hypothetical case that would imply that "infant-colonizing maternal microbes" (l. 220) where neither under positive nor negative selection, even though some species were completely selected out (i.e. strong negative selection on them). I think underlying this problem is that changing the pool conflates selection of which species are actually "allowed" in the regional pool and selection going on within the local communities.

Don't get me wrong, I find this genuinely interesting and appreciate the authors including different regional pools in their analysis, while at the same time I think simply swapping the species pool in Sloan's model (or others, for that matter) might conceptually not be as trivial as they make it sound.

Long story short, please discuss the implications of changing the regional species pool, why it should not be a "big" problem in their study (large overlap between infant and mother pools?) and hint at possible problems with the interpretation of the results coming from this.

Reviewer #2 (Remarks to the Author):

Sprockett et al. present a unique dataset containing longitudinal gut and oral microbiota samples from numerous mother-infant pairs from several Tsmaine villages in Bolivia. Using these data,

they investigate the level of bacterial sharing between parent and child across body sites finding several interesting patterns including associations between microbiome composition and dietary habits. They then extend their analyses using a neutral community model to infer the relative contribution of neutral and selective processes to infant microbiome colonisation in the mouth and gut, further extending this comparison to see how these processes are manifest in Bangladeshi and Finnish populations. Finally, to infer the processes driving colonisation in the Tsmaine groups, they explore the effects of modernisation on microbial community structures and dynamics by contrasting more and less isolated villages.

Overall, I think this is a dataset of incredible value and think the initial analyses showing the relative overlaps and metadata contributions to microbiome effects are of great interest. The figures present the data very nicely and add much value to the manuscript. The data processing approaches applied to the microbiome data also seem very appropriate.

However, I think there is a lack of statistical detail in the methods section with many specific details isolated to figure legends or missing. I also have two main concerns regarding the analyses. The first is the suitability of the background pools used to carry out the neutral modelling (see point 8) and the second is the use of UK adults to normalise the Finnish infant data (point 10). I must state that I am no expert in the first matter but have commented based on my current understanding – a more familiar reviewer may be able to clarify on this matter more clearly. Please find more specific comments regarding the manuscript below.

1. Abstract: How are there 52 mother-infant pairs if there are only 48 infants and 51 mothers?

2. Figure 1 Panel a) unclear what the y axis represents, assume each line is a dydad. b) I was unclear how a mixed effects model with multiple variables including random effects is represented in a single linear line.

3. Line 118: How was the PERMANOVA to determine the contribution of metadata to microbiome composition performed? Was it carried out on a per-variable basis or using a combined model of multiple variables? If the latter, was the order permuted? As PERMANOVAs are sensitive to the order of variables in a model. If the former, how many variables were tested, was multiple testing correction used, and can the other results be included in the manuscript?

4. Line 125: Is there any reason that these results might differ from the previously published study?

5. Line 145: Were the coefficients for age and MDD association estimated from individual models or one combining both as predictors? Further information could be added to the supplementary figure 2 legend and/or methods.

6. Line 160: This seems like one of the most important take-aways from the paper, the direct overlap between individual mother-child pairings. Could you consider including the actual percentages and standard deviations for these overlaps here. The increased uniqueness of the child stool compared to the oral microbiome is particularly interesting.

7. Line 164: Another interesting way to look at the overlap between mother to child vs. the rest of the village would be to compare the beta-diversity distances between mother-child pairs and mother non-child pairs (for stool and oral individually). This would give an idea of how much more similar they are than just random pairings. (Just an idea!)

8. Line 188 onwards, section on Neutral Selection Modelling.

I would preface this comment by again highlighting that I am by no means an expert in the modelling approach that has been applied here, but from reviewing the Sloan et al. reference cited

in the manuscript, my understanding is that, given an underlying global pool of possible species, it is possible to infer if the observed species distributions are more driven by stochastic immigration and duplication events versus being driven by active selection processes. Therefore, the results of the models will be highly dependent on how the background pool is defined. In this section, two pools are used - one using microbes from the infants and a separate one from the mothers.

I wonder if in the first case there is an overestimation of the contribution of neutral processes, as there have already been selection processes that determined the ability of these taxa to enter and colonise the infants. Similarly, the other mother-based pool might over estimate selection, as it will be enriched for microbes that are only found in the mothers and cannot pass into the infants as they are always selected against.

Could the same models also be applied but using all of the microbes observed across all samples (all ages, both oral and stool) as the universal background pool? The assumption then being that the ability to pass from the mother to the baby (or be taken up from the environment) and then stably colonise is a mode of selection. This seems like it would more reflect the aims at the start of this section that "ICABs may increase the opportunity for neutral dispersal of microbes from mothers to their children".

If this is indeed the case, then choosing a pool only containing species within the 3-month window used in the models in the section starting at line 259 might also be over estimating neutral processes. As the loss or gain of taxa overtime could also be due to changing selective processes, but such taxa would be excluded by limiting the window of the pool.

9. Line 248 and Fig 4a: Jaccard distances will not take into account the abundance or phylogenetic structures in the datasets, whilst the overlaps of ASVs might be similar, the global community composition might be very different. What would this plot look like using a weighted UniFrac distance for instance? This might uncover some divergences related to comment 11. Indeed, UniFrac distances would arguably be more informative throughout.

10. I am unsure that the UK adult population is a suitable comparison for the Finnish infants. Looking at Supplementary Figure 4 the only significant difference in maturation rate is with the European samples. This may indeed be due to underlying differences in this group; however, it is also the only one using a non-matching adult population for the normalisation. For instance, *Prevotella* is almost absent in the Finnish samples but the adult UK population contains the expected *Prevotella* rich enterotype group. Differences have been observed between Dutch, Flemish, US and UK adults (e.g. PMID: 27126039). Suggesting that the UK population might be even more distinct to the Finnish than other European datasets. I think using the current approach, it is not possible to ascertain that differences in the European dataset are not due to technical limitations.

11. The PCA in figure 5b uses Bray-Curtis distances. Is there a rationale for swapping between different ordination approaches throughout the manuscript? It is not always clear which is used either (for instance Fig 1d.)

12. Line 343. Is the Christensenellaceae subnetwork associated with BMI in the Tsimane data?

13. Line 362: The statement that the Tsimane display a distinct maturation trajectory may be overstated. In terms of rate, it was almost identical to the Bangladeshi data and I think that the trajectories inferred from the PCA in Fig. 4a may be less distinct using an abundance-based metric for ordination and may be misleading given the inclusion of the UK adults (as in point 10).

14. Line 392. I think this is a nice take away from the manuscript but might also be worth adding that, additionally, multi-generational dietary changes in the developing village may also contribute to the observed shifts. The work from the Sonnenburg lab is a nice reference for this.

Reviewer #3 (Remarks to the Author):

Summary

This study represents an important examination of maternal and fecal and oral microbiomes among the Tsimane of Bolivia. It was well-written manuscript that provided novel and interesting insight into the relationships between mother and infant microbial communities, diet, and geographical location and how neutral or selective processes can describe unique taxa found in oral and fecal samples of the dyad. Having paired samples from mother-infant dyads over time adds increased importance to the paper. This manuscript also provides important insights into maternal-infant microbiomes in non-Western, small-scale developing societies, essential data to furthering our understanding of human microbiomes. The application of a neutral community model (NMC) of assembly proved useful in assessing which bacteria may be selectively assembled and/or excluded.

General

The authors flip back and forth throughout the manuscript with using "gut," "fecal," and "stool" microbiome/microbiota. I do not recommend using the term "gut" as it is colloquial and nonspecific when it comes to the human gastrointestinal tract. In addition, the research team did not collect intestinal samples, which is implied when they state "distal gut samples."

There are several instances (e.g., line 99) where it is recommended that the authors replace causal verb such as "affect" with phrases such as "are associated with."

A table or figures (stacked bar charts) characterizing the relative amounts of various bacterial taxa in each of the site for both mothers and infants by village would be helpful early in the results.

The "global" analysis of Finnish and Bangladeshi may be better presented in a separate paper. It made the paper feel disjointed and pulled the reader away from the main results. Those analyses are also complicated by the different collection methods, protocols, pipelines, etc. which make the results more difficult to interpret. I have considerable concern related to variation in methods used for DNA extraction in the "global" data (see additional comments below).

Results

Line 102-103: There appears to be a discrepancy in the overall sample numbers. In the materials and methods, it says "focused on 52 mother-child dyads"; whereas, here it says 48 infants and 51 mothers.

Line 104: These sentences seem more like materials and methods. A table that describes the anthropometric measures and some of the health and diet survey data from these dyads would be helpful in providing insight into other variables that may influence the microbial compositions. This information may be in one of the two articles cited but it would be helpful to the reader to not have to go through both of these other manuscripts to find the information relevant to these dyads.

Line 106: Also, the reference here is [8]; whereas, in the materials and methods it is [9]. Why the different citations?

Line 114: Does this reference [28] support that maternal stool samples decrease in diversity? I do not think it does. I don't think they saw changes in the Shannon index post delivery. In another

study (Carrothers et al. 2015), they also did not see changes over time during the first 6 months postpartum.

Line 120-121: What was the unit of measure when comparing villages? Was it village or individual?

Line 125-126: Figure 1d – Perhaps I’m missing it, but I don’t see how this shows that tongue swabs of 16-18 month olds harbor microbial communities...

Line 133: How can there be a high prevalence of stunting and underweight but low prevalence of undernutrition (need to read reference 30).

Line 151: How is this “in contrast to dietary factors?”

Line 156: Please consider adding “Potential” before “Sources”

Line 180: This is only in the those shared with the infant’s stool and tongue swab, correct?

Line 214: The authors state the ASVs in the infant stool fit the neutral model moderately well. But the value is closer to 0 than 1. Wouldn’t one have expected the R² value to be closer to 1 or at least above 0.5 to say that it fit the neutral model moderately well?

Line 216: I would be interested in which taxa experienced positive selection. These would likely be “beneficial” in some way, and it might be useful to know which ones were included in this group.

Line 219: When the mother pool was used, it was a poorer model fit (R² = -0.29). The model fit for mother tongue was 0.19 and the authors have indicated that this still fit the neutral model. Is an R² above 0.1 considered a relatively good fit that indicates neutral dispersal; whereas, values that are very close to 0 or negative, indicative of either positive or negative selection? Additionally, in Line 229, I would have thought that an R² of 0.19 might also be indicative of selection. Perhaps I am just misunderstanding this.

Just curious if these analyses were separated out by groups of infant age, I wonder how the R² values might change?

Line 228: From my viewpoint, I wouldn’t say that the model still fit “relatively well” for the mother’s tongue swab pool.

Line 286+: Although it’s interesting to speculate that market access is a driving force behind variation in microbial community structures, there is a plethora of other differences among these villages. I would be extremely careful here to suggest these differences are due to market access or market integration. Can you further refine these analyses to take into account some of the other differences between these communities?

Line 293-295: Which sites have the higher intake and higher levels? I am guessing that these are referring to the proximal river sites?

Line 300-302: Might season influence the outcome of differences between village ecotypes? Were samples treated the same from collection through sample processing? Providing a statement in this regard would be helpful to understand the potential factors that might be at play in influencing the microbial composition.

Line 364-367: What is the “same body site” that is being referred to in this sentence? Not sure the ref supports the statement that “other maternal body sites is relatively rare.” The Ferritti paper describes contributions from maternal vaginal, skin, oral, and gut, with maternal vaginal and skin contribution decreasing during the first few days postpartum.

Line 367-370: Understanding the contribution of ICABs on infant microbial assembly, structure, and dynamics is very intriguing. Are there any data that can be presented to support that levels of various ICABs are related to microbial composition?

Materials and Methods

Line 412-413: Please provide your UCSB IRB protocol #s.

Line 423: Please provide a further explanation of what mixed longitudinal design means.

Was the longitudinal sampling spread across all 6 sites or for example, were the samples collected during early months after birth collected from one site and later after birth collected from another site?

Line 433+: Please provide additional information related to the sample collection containers, including whether they were sterile. What instructions were the subjects given in terms of how to collect their samples? Were samples collected from the ground? Were all samples treated the same? Also, additional information is needed as to how the fecal samples were transferred into the containers and whether there was any sort of cleansing step prior to this.

Line 440: Please provide additional information regarding what beads were used.

Line 443: What is an error-correcting barcode?

Line 451:

Current study – DNA extracted using MoBio PowerSoil-htp kit with bead-beating; V4 region using 515F and 806R

Bangladesh (Subramanian et al. 2014) – DNA isolated by beadbeating in phenol/chloroform, purified with QIAquick column. Not totally clear what positions the V4 primers were.

Finland (Vatanen et al. 2016) – DNA isolated using the QIAamp DNA Stool Mini Kit. PCR following Caporaso et al. 2012)

England – (Goodrich et al. 2016) – unclear how DNA isolated; references Goodrich et al. 2014 but no details here. Amplified V4 using primers 515F and 806R); sequenced on MiSeq 2x250 bp paired-end sequencing

Although all 3 studies sequenced the V4 region, there were differences in the DNA extraction procedure. It is fairly well documented that different DNA extraction methods can yield different results; thus, although the authors state that methods were relatively similar, caution and/or mention of this as a limitation to these comparisons should be mentioned.

Were there any negative controls done with the extraction and PCR and if so, were they sequenced? If so, did the authors consider running decontam to identify potential contaminants?

Were any positive controls (e.g. mock communities?) extracted and/or was the DNA amplified with PCR? If so, were they sequenced?

Line 455: Why were Archaea included?

Line 471: This is the first mention of the MDD food groups. There is a nice description of this FFQ in the results section, but I would have expected it to be included in the methods section (most people still read the methods before they read the results).

Line 496: Similarly, I would have appreciated a more extensive discussion of the "regional species pool" in the methods section rather than the results section.

FIGURES

Figure 1 Not sure what the y axis is on Figure 1a. Text mentions statistical outcomes for Figure 1e, and it would be nice if they were also on the figure.

Figure 1e: What do the circles represent? Are these ASVs or are they infant stool taxa composition of various samples?

Figure 2: In order to be counted as shared, was the ASV shared between at least one individual mother/infant dyad or multiple mother/infant dyads?

Or shared between any mother sample and infant sample?

And were they compared across all samples from the mother / infant dyad? Or just the matched timepoint?

Figure 3 I would recommend splitting this figure into at least 2 separate figures. It's next to impossible to read the ASV/taxa designations.

Figure 3a: Would it be possible to make the color for the "Below prediction" closer to the row color in 3b? This might help in relating the two together.

Figure 3b. I wonder since the authors are specifying specific ASVs, could they highlight these somehow either with an arrow or a square around the circle?

Figure 4a: Would help to color the Finnish infants and English adults a different color. And maybe even include a different shape?

Figure 4a: What do the lines linking the samples represent?

Figure 4e: Once again, font is small.

Figure 5 Can the authors make the symbols in Figure 5a the same as those in Figure 5b?

Figure 5d: Font of ASV numbers are very small.

Figure 6a: What does the length of the edges between nodes represent?

Supplementary Figure 3c: Missing 'Left panel – infant stools, right panel – infant tongue swabs'

Supplementary Figure 5: The shape of the curve is different among the different populations. What, if any, is the significance of this?

Reviewer #4 (Remarks to the Author):

This manuscript reports the results of a longitudinal study of the oral and gut microbiomes of 52 infant-mother pairs in an indigenous Brazilian population. Lifestyles of the Tsimane people is undergoing change including increased access to market good and medicine, factors that have

direct potential to influence their microbiomes. The manuscript is written well and will be of most interest to microbiome researchers, but is accessible to a broader scientific audience.

The authors focus on infant care associated behaviors that are likely to increase maternal to infant transmission. In particular, pre-masticating foods and preparation of chicha – a fermented drink inoculated with saliva – have the obvious potential to influence colonization of the infant gut. Results are also compared with microbiota from infants in Bangladesh and Finland. The conclusion that changes in lifestyle are affecting the trajectory of the gut microbiome is intriguing and of broad interest. Many of the other findings confirm published results and extend them to the Tsimane population.

The major concern I have is in the interpretation and presentation of the neutral model. While application of the neutral model is appropriate for this study, the interpretation of results from the neutral model, namely that "...the majority of microbes colonizing infants from different countries were neutrally distributed" could easily be misinterpreted by readers. The neutral model makes the provocative assertion that all individuals, in this case amplicon sequence variants (ASVs) are ecologically identical and that their occurrence in the microbiome is based solely on the pool of ASVs available to colonize the gut or mouth. A neutral distribution in this case does not mean that there is no selection, but rather that selection in the larger pool is the same as selection in the target population. The gut is a strongly selective environment and that may be missed in the current presentation.

Responses to Reviewers

Reviewer #1:

The study explores the microbiota of infants and their mothers from an indigenous Bolivian population, with a particular emphasis on the contribution of neutral processes to the assembly of the microbiota. I find the study generally interesting, in particular because it includes both a temporal and spatial view on the changes in the microbiota of the sampled individuals.

I have however two major questions regarding the application of the neutral model, which seem central to their interpretations, and which I would like to see addressed before publication.

1) What's the exact interpretation of the R^2 values given throughout the paper? As generalized R^2 values for non-linear fits do not allow for a direct interpretation as "variance explained" and are not bounded between 0 and 1, I'm wondering what a value of e.g. $R^2 = 0.38$ actually "means", other than the model fits the data "moderately well" (l. 214)? This is in particular an issue when comparing goodness of fit across different datasets and drawing central conclusions like "market integration may increase the role of selection" (l. 312) because R^2 "decreased in samples with higher levels of market access" (l. 311 and Fig 5c). Looking at the fits and R^2 I can only see that the model fits "moderately well" in all cases, but a more quantitative interpretation of the R^2 values seems difficult. Or in other words, does $R^2=0.45$ (proximal river) really indicate a significantly "less well" fit than 0.61 (forest), given that these values do not correspond to actual variance explained?

Response:

We thank the reviewer for this useful feedback, and have worked to address these points in several ways. First, it is true that the R^2 values calculated in this analysis do not allow for a straightforward quantitative interpretation of variance explained, since the model is non-linear. While it does provide a more qualitative measure of goodness-of-fit (numbers closer to 1 indicate a better fit), we have instead decided to report the root mean squared error (RMSE) to assess the fit of the Neutral Community Model (NCM) to prevent readers from erroneously interpreting the R^2 values as variation explained. RMSE is very similar to the R^2 calculated before, and provides a similar interpretation, although in this case, lower values indicate a better fit (0 would indicate a perfect fit) (see new text on page 9). Second, we have removed subjective statements such as a model fits 'moderately well', and instead only report quantitative values. Third, in cases where we compare NCM fit between two different cohorts such as between the different villages types, we now subsample the datasets, one sample per subject, with 1000 permutations. This generates a distribution of bootstrapped RMSE values, and allows us to

assess better our confidence in the mean. We believe that these revisions allow us to provide a more meaningful description of model fit, as well as a more robust assessment of differences in NCM fit between groups.

2) I find the approach of using different regional species pools (infant and mother pools) very interesting. But I have some problems with the conceptual interpretation of this.

Imagine for example that the mother pool is largely identical to the infant pool in terms of relative abundances, but contains a few additional species which are not present in infants. These species are dropped from the analysis (if I read the accompanying R code correctly?), which makes sense I guess. But because otherwise the pools are almost identical, I would expect the fit to the species actually present in infants to not change very much. By the authors logic, if I understood it correctly, in this hypothetical case that would imply that “infant-colonizing maternal microbes” (l. 220) where neither under positive nor negative selection, even though some species were completely selected out (i.e. strong negative selection on them). I think underlying this problem is that changing the pool conflates selection of which species are actually “allowed” in the regional pool and selection going on within the local communities.

Don't get me wrong, I find this genuinely interesting and appreciate the authors including different regional pools in their analysis, while at the same time I think simply swapping the species pool in Sloan's model (or others, for that matter) might conceptually not be as trivial as they make it sound.

Long story short, please discuss the implications of changing the regional species pool, why it should not be a “big” problem in their study (large overlap between infant and mother pools?) and hint at possible problems with the interpretation of the results coming from this.

Response:

We appreciate the thoughtful assessment of our analysis, and in response have made significant updates to our analytical framework and discussion thereof.

First, the reviewer is correct in stating that bacterial taxa found in the mother but not the infant gut are dropped from the analysis. In that sense, they would not be “infant-colonizing maternal microbes” (l. 220), since they do not colonize the infant. However, to emphasize that point to the reader, we added these non-infant-colonizing maternal microbes to the main text, as well as to Fig. 3a and Supplementary Fig. 7a, explicitly mentioning that they are not able to colonize the infant gut either due to dispersal limitation or negative selection.

However, we concur that interpreting the results when changing the make-up of the regional species pool is not trivial, and have therefore removed this element of the analysis. Sloan *et al.* (2006) originally formulated the NCM using values for source community abundance that were inferred by averaging local

observed abundances in the samples, specifically in contrast to previous models that required more specific knowledge about metacommunity dynamics and evolution. We agree that it is preferable to use the NCM as it was originally formulated.

Reviewer #2:

Sprockett et al. present a unique dataset containing longitudinal gut and oral microbiota samples from numerous mother-infant pairs from several Tsmaine villages in Bolivia. Using these data, they investigate the level of bacterial sharing between parent and child across body sites finding several interesting patterns including associations between microbiome composition and dietary habits. They then extend their analyses using a neutral community model to infer the relative contribution of neutral and selective processes to infant microbiome colonisation in the mouth and gut, further extending this comparison to see how these processes are manifest in Bangladeshi and Finnish populations. Finally, to infer the processes driving colonisation in the Tsmaine groups, they explore the effects of modernisation on microbial community structures and dynamics by contrasting more and less isolated villages.

Overall, I think this is a dataset of incredible value and think the initial analyses showing the relative overlaps and metadata contributions to microbiome effects are of great interest. The figures present the data very nicely and add much value to the manuscript. The data processing approaches applied to the microbiome data also seem very appropriate.

However, I think there is a lack of statistical detail in the methods section with many specific details isolated to figure legends or missing. I also have two main concerns regarding the analyses. The first is the suitability of the background pools used to carry out the neutral modelling (see point 8) and the second is the use of UK adults to normalise the Finnish infant data (point 10). I must state that I am no expert in the first matter but have commented based on my current understanding – a more familiar reviewer may be able to clarify on this matter more clearly. Please find more specific comments regarding the manuscript below.

1. Abstract: How are there 52 mother-infant pairs if there are only 48 infants and 51 mothers?

Response:

We agree that this is not clearly described. Enrolled individuals were assigned to dyads (i.e., given a unique family ID) even if other family members did not enroll. If we only count dyads with samples from both the mother and infant, then there were only 47 where at least one sample was collected from each member of the pair. The remaining singleton samples, (i.e., samples from only the infant or the mother of a dyad) have been removed from the analysis; this is now described in the Materials and Methods. This point has been clarified

throughout the manuscript. Some p-values and R^2 values have changed slightly with these samples now excluded.

The text now reads: “The first set of samples presented in this manuscript was collected between September 2012 and March 2013 using a mixed longitudinal design, and focused on mother-child dyads. A total of 156 stool samples and 58 tongue swabs were collected from 48 infants, and 134 stool samples and 62 tongue swabs were collected from 51 adults (see Supplementary Table 1). Samples from incomplete dyads (i.e., dyads from which samples were available only from the infant, or only from the mother) were excluded from further analysis. The remaining 47 infant-mother dyads resided in six villages along the Maniqui River.

2. Figure 1 Panel a) unclear what the y axis represents, assume each line is a dyad.

Response:

We have updated the y-axis label in Figure 1a to read “Dyad”.

b) I was unclear how a mixed effects model with multiple variables including random effects is represented in a single linear line.

Response:

Figures 1b and 1c were based on a linear mixed-effects (LME) model to regress the single variable, “Shannon Diversity Index” against time-since-delivery, independently for infants and their mothers and for each of two body sites. Our goal was to evaluate temporal trends in the diversity of the bacterial communities of both body sites during the postpartum period. However, since multiple samples were collected from each individual, we accounted for the subject-structure of these longitudinal data by treating subject as a random effect. The red (panel b) and blue (panel c) lines indicate the linear mixed-effects regression of diversity on time with grouping by subject. This approach is similar to that used in DiGiulio *et al.* Fig. 1a (PNAS 2015; 112:11060-5). [Note: The p-values and R^2 values have been updated to reflect re-analysis without the data from samples from incomplete dyads, as highlighted in our response to Comment #1 above.]

3. Line 118: How was the PERMANOVA to determine the contribution of metadata to microbiome composition performed? Was it carried out on a per-variable basis or using a combined model of multiple variables? If the latter, was the order permuted? As PERMANOVAs are sensitive to the order of variables in a model. If the former, how many variables were tested, was multiple testing correction used, and can the other results be included in the manuscript?

Response:

The PERMANOVA for determining the contribution of metadata to microbiome composition was performed using a combined model of multiple variables.

The variables we used were “body site” (stool or tongue swab), “age class” (adult or infant), and “village type” (proximal or distal river village). Our first objective with this analysis was to quantify how much of the variation in the dataset could be attributed to age and body site (related to Fig. 1d). We also included village type here because later in the manuscript we examined the effect of this variable on infant microbiome assembly, and thought it was pertinent to set up that analysis here. Since the terms were added sequentially from first to last, we permuted their order and found that the order had a very small effect on the results (<0.2% of the variance and essentially no change in p-values). All variables accounted for a significant fraction of the variance; we report the mean of the variance and p-value permutations in the manuscript.

4. Line 125: Is there any reason that these results might differ from the previously published study?

Response:

With additional analysis, we have confirmed that our results do not differ with those of the previous study, and in fact support the previous finding of Han *et al.* (2016). To provide more details of our analysis, we include a new supplementary figure (Supplementary Fig. 3) with data that are more directly comparable to the analysis of Han *et al.* (Fig. 3).

5. Line 145: Were the coefficients for age and MDD association estimated from individual models or one combining both as predictors? Further information could be added to the supplementary figure 2 legend and/or methods.

Response:

The coefficients were estimated from a single model that used both Total_Diet_Diversity and Age_Months as predictors. The main text has been updated to make this more clear, and now reads:

“A single linear mixed-effects model using MDD and age as predictors showed that infants’ MDD scores were positively correlated with infant stool microbiota diversity (Supplementary Fig. 5a, $p = 0.032$, conditional $R^2 = 0.315$), and accounted for a larger effect than age ($\beta_{\text{MDD}} = 1.2$ vs. $\beta_{\text{Age (months)}} = 0.6$, $p < 0.05$ for both) on infant stool diversity.”

6. Line 160: This seems like one of the most important take-aways from the paper, the direct overlap between individual mother-child pairings. Could you consider including the actual percentages and standard deviations for these overlaps here. The increased uniqueness of the child stool compared to the oral microbiome is particularly interesting.

Response:

We appreciate this suggestion. The updated text now reads,

“On average, 46.2% of the microbes colonizing an infant’s gut or tongue were also found in their mother’s samples from the same body site. This percentage increased in stool samples as the infant became older (Supplementary Fig. 6a, blue line plus green line), increasing from 38.4% in 0-6 month olds to 50.5% in 12-18 month olds (Wilcoxon rank-sum test, $p = 0.0013$); in tongue swabs, the increase from 59.6% in 0-6 month olds to 63.8% in 12-18 month olds was not statistically significant (Wilcoxon rank-sum test, $p = 0.52$).”

7. Line 164: Another interesting way to look at the overlap between mother to child vs. the rest of the village would be to compare the beta-diversity distances between mother-child pairs and mother non-child pairs (for stool and oral individually). This would give an idea of how much more similar they are than just random pairings. (Just an idea!)

Response:

We agree. We have added this comparison as Supplementary Figure 3.

8. Line 188 onwards, section on Neutral Selection Modelling.

I would preface this comment by again highlighting that I am by no means an expert in the modelling approach that has been applied here, but from reviewing the Sloan et al. reference cited in the manuscript, my understanding is that, given an underlying global pool of possible species, it is possible to infer if the observed species distributions are more driven by stochastic immigration and duplication events versus being driven by active selection processes. Therefore, the results of the models will be highly dependent on how the background pool is defined. In this section, two pools are used - one using microbes from the infants and a separate one from the mothers.

I wonder if in the first case there is an overestimation of the contribution of neutral processes, as there have already been selection processes that determined the ability of these taxa to enter and colonise the infants. Similarly, the other mother-based pool might over estimate selection, as it will be enriched for microbes that are only found in the mothers and cannot pass into the infants as they are always selected against.

Could the same models also be applied but using all of the microbes observed across all samples (all ages, both oral and stool) as the universal background pool? The assumption then being that the ability to pass from the mother to the baby (or be taken up from the environment) and then stably colonize is a mode of selection. This seems like it would more reflect the aims at the start of this section that “ICABs may increase the opportunity for neutral dispersal of microbes from mothers to their children”.

If this is indeed the case, then choosing a pool only containing species within the 3-month window used in the models in the section starting at line 259 might also be over estimating neutral processes. As the loss or gain of taxa overtime could also be due to changing selective processes, but such taxa would be excluded by limiting the window of the pool.

Response:

We thank the reviewer for their thoughtful assessment of our analysis. We have made significant updates to our analysis in response to various reviewer comments. With regard to the reviewer's first comment, we have made updates to the text and added Fig. 3a and Supplementary Fig 7a to make it more clear that bacterial taxa in the mother that do not colonize the infant are not considered in the NCM, and are potentially under negative selection or dispersal limitation. We have also removed the analysis that considers the fit of the model using the maternal species pool, largely because the interpretation is non-trivial. It conflates our direct estimate of the regional species pool, as originally formulated by Sloan *et al.* (2006), with our assumptions about its structure and composition.

Regarding the reviewer's comment about aggregating all infant and maternal samples as a universal background pool, while one could do this, there are inherent assumptions in doing so that would make interpretation difficult and unclear. In particular, it would assume that each of these potential sources contributes equally to the regional source pool, which is unlikely to be the case. By limiting our analysis to its most straightforward formulation (where the source pool is approximated by the average of the local communities), we avoid such complications because this approach does not make any specific assumptions about what environments make up the source pool; it simply assumes that whatever that source pool may be, it can be approximated by averaging the local communities.

9. Line 248 and Fig 4a: Jaccard distances will not take into account the abundance or phylogenetic structures in the datasets, whilst the overlaps of ASVs might be similar, the global community composition might be very different. What would this plot look like using a weighted UniFrac distance for instance? This might uncover some divergences related to comment 11. Indeed, UniFrac distances would arguably be more informative throughout.

Response:

We have added Supplementary Fig. 8, which displays the data in Fig. 4a using two non-phylogenetic distance metrics, Jaccard similarity index and Bray-Curtis distance, and both unweighted and weighted Unifrac.

10. I am unsure that the UK adult population is a suitable comparison for the Finnish infants. Looking at Supplementary Figure 4 the only significant difference in maturation rate is with the European samples. This may indeed be due to underlying differences in this group; however, it is also the only one using a non-matching adult population for the normalisation. For instance, *Prevotella* is almost absent in the Finnish samples but the adult UK population contains the expected *Prevotella* rich enterotype group. Differences have been observed between Dutch, Flemish, US and UK adults (e.g. PMID: 27126039). Suggesting that the UK population might be even more distinct to the

Finnish than other European datasets. I think using the current approach, it is not possible to ascertain that differences in the European dataset are not due to technical limitations.

Response:

The reviewer's point is a good one. The UK adult population has been removed from the analysis.

11. The PCA in figure 5b uses Bray-Curtis distances. Is there a rationale for swapping between different ordination approaches throughout the manuscript? It is not always clear which is used either (for instance Fig 1d.)

Response:

Throughout the manuscript, we used the Jaccard Index consistently when we were interested in exactly which microbes were shared between two communities, as in Fig. 1d. Fig. 5b did not specifically examine sharing; instead, our goal was to describe more general patterns in the adult stool microbiota data.

However, in re-considering the question at hand, we now believe that the plot of PC1 from a PCoA using Bray-Curtis distance, versus Shannon diversity index, previously shown in Supplementary Fig. 6a, is more relevant and effective; hence, we now present the original Supplementary Fig. 6a as the new Fig. 5b, and the original Fig. 5b (PCoA based on Bray-Curtis) as the new Supplementary Fig. 10d. Supplementary Fig. 10 now also presents PCoA plots based on weighted and unweighted Unifrac distances, and the Jaccard similarity index.

In addition, the legend for Fig. 1d has been updated to specify that the PCoA is based on the Jaccard Index of stool and oral swab samples.

12. Line 343. Is the Christensenellaceae subnetwork associated with BMI in the Tsimane data?

Response:

No, it is not.

13. Line 362: The statement that the Tsimane display a distinct maturation trajectory may be overstated. In terms of rate, it was almost identical to the Bangladeshi data and I think that the trajectories inferred from the PCA in Fig. 4a may be less distinct using an abundance-based metric for ordination and may be misleading given the inclusion of the UK adults (as in point 10).

Response:

The analysis of the maturational trajectory of each population has been removed from the manuscript.

14. Line 392. I think this is a nice take away from the manuscript but might also be worth adding that, additionally, multi-generational dietary changes in the developing village may also contribute to the observed shifts. The work from the Sonnenburg lab is a nice reference for this.

Response:

While we agree that the work from the Sonnenburg Lab is an important finding, showing that dietary shifts compound the loss of diversity over generations (Sonnenburg *et al.* 2016, *Nature*), the current study was completed over the course of only two field seasons. We have no evidence that suggests that multi-generational dietary changes contribute to the microbiome changes that we observed, since that would require observations over several generations.

Reviewer #3 (Remarks to the Author):

Summary

This study represents an important examination of maternal and fecal and oral microbiomes among the Tsimane of Bolivia. It was well-written manuscript that provided novel and interesting insight into the relationships between mother and infant microbial communities, diet, and geographical location and how neutral or selective processes can describe unique taxa found in oral and fecal samples of the dyad. Having paired samples from mother-infant dyads over time adds increased importance to the paper. This manuscript also provides important insights into maternal-infant microbiomes in non-Western, small-scale developing societies, essential data to furthering our understanding of human microbiomes. The application of a neutral community model (NMC) of assembly proved useful in assessing which bacteria may be selectively assembled and/or excluded.

General

→ The authors flip back and forth throughout the manuscript with using “gut,” “fecal,” and “stool” microbiome/microbiota. I do not recommend using the term “gut” as it is colloquial and nonspecific when it comes to the human gastrointestinal tract. In addition, the research team did not collect intestinal samples, which is implied when they state “distal gut samples.”

Response:

For the sake of consistency, all uses of “feces” have been changed to “stool” throughout the manuscript. “gut microbiota” has been changed, where appropriate, to “stool microbiota”. However, there are a few references to the gastrointestinal tract that seem most appropriate.

→ There are several instances (e.g., line 99) where it is recommended that the authors replace causal verb such as “affect” with phrases such as “are associated with.”

Response:

The verb “affect” has been replaced with “are associated with” where appropriate in the main text, including line 113 (formerly line 99).

→ A table or figures (stacked bar charts) characterizing the relative amounts of various bacterial taxa in each of the site for both mothers and infants by village would be helpful early in the results.

Response:

Stacked bar charts are now included (Supplementary Figures 1 and 2).

→ The “global” analysis of Finnish and Bangladeshi may be better presented in a separate paper. It made the paper feel disjointed and pulled the reader away from the main results. Those analyses are also complicated by the different collection methods, protocols, pipelines, etc. which make the results more difficult to interpret. I have considerable concern related to variation in methods used for DNA extraction in the “global” data (see additional comments below).

Response:

We understand the reviewer’s point, yet we feel that this comparative analysis is important in providing context for our analysis of microbiota assembly in the Tsimane. Since the ICABs that serve as a focus in our investigation are essentially universal among Tsimane mothers, it is necessary for us to look elsewhere at other societies with distinct ICABS.

Results

→ Line 102-103: There appears to be a discrepancy in the overall sample numbers. In the materials and methods, it says “focused on 52 mother-child dyads”; whereas, here it says 48 infants and 51 mothers.

Response:

We agree that dyad, subject, and sample numbers were not clearly explained, and have updated the manuscript and analyses accordingly. See response to Reviewer 2, comment 1.

Briefly, enrolled individuals were assigned to dyads (i.e., given a unique family ID) even if other family members did not enroll. This is how we were left with

more individual infants and adults than dyad pairs. If we only count dyads where at least one sample was collected from both the mother and infant, then there were 47 “complete” dyads. To clarify this point, the remaining singleton samples, (i.e., samples from only the infant or the mother of a dyad) have been removed from downstream analyses.

→ Line 104: These sentences seem more like materials and methods. A table that describes the anthropometric measures and some of the health and diet survey data from these dyads would be helpful in providing insight into other variables that may influence the microbial compositions. This information may be in one of the two articles cited but it would be helpful to the reader to not have to go through both of these other manuscripts to find the information relevant to these dyads.

Response:

The information contained in the last two sentences of this paragraph is also found in the Methods section; hence, these two sentences have been removed. In addition, Supplementary Table 2 has been added to the manuscript which details all of the measures and survey data used in the analysis. The data were previously available in the code repository, but they are now provided in this Supplementary Table as well, so that they are more accessible to readers.

→ Line 106: Also, the reference here is [8]; whereas, in the materials and methods it is [9]. Why the different citations?

Response:

Citation [8] is a review of all of the manuscripts that have been published by the Tsimane Health and Life History Project, and citation [9] is one of those manuscripts. The current data were specifically collected for citation [9], which should be cited in both places. Those references have now been corrected in response to the previous comment.

→ Line 114: Does this reference [28] support that maternal stool samples decrease in diversity? I do not think it does. I don't think they saw changes in the Shannon index post delivery. In another study (Carrothers et al. 2015), they also did not see changes over time during the first 6 mo postpartum.

Response:

The reviewer is correct that DiGiulio *et al.* (2015) did not observe a decrease in maternal stool diversity in the post-partum period. In fact, after we removed the samples from mothers without a paired enrolled infant (see previous comment), our marginally significant result ($p = 0.04$) became not significant (0.062). Thus, we have removed the entire statement from the manuscript.

→ Line 120-121: What was the unit of measure when comparing villages? Was it village

or individual?

Response:

The type of village in which the infant lived (proximal or distal to the regional market) explained <0.5% of the variation in the infant and maternal samples. If we understand the question correctly, the comparison was made at the level of individual.

→ Line 125-126: Figure 1d – Perhaps I'm missing it, but I don't see how this shows that tongue swabs of 16-18 month olds harbor microbial communities...

Response:

Figure 1d shows that the microbial communities from the tongue swabs of 16-18 month old infants cluster with those from the tongue swabs of the mothers, indicating that the two age groups harbor communities of similar composition at that body site. This is in contrast to the stool samples from 16-18 month old infants, most of which do not cluster with the maternal stool samples, indicating that they harbor distinct microbial communities.

→ Line 133: How can there be a high prevalence of stunting and underweight but low prevalence of undernutrition (need to read reference 30).

Response:

This text has now been revised to clarify that growth faltering might be due to pathogen burden, rather than nutritional deficiencies:

Growth faltering in the Tsimane relative to international standards might be due to pathogen burden, rather than lack of nutrition, as indicated by a much higher prevalence of height-for-age z-scores (HAZ) < -2 SD (53%), than weight-for-age (WAZ, 16%) or weight-for-height z-scores (WHZ 9%) < -2 SD in children aged 2-5 years.

→ Line 151: How is this "in contrast to dietary factors?"

Response:

Our phrasing was unclear, since it was originally meant to contrast the direction of the effect, not to contrast the sources of the effect. It has now been removed from the manuscript.

→ Line 156: Please consider adding "Potential" before "Sources"

Response:

The word "potential" has been added, as recommended.

→ Line 180: This is only in the those shared with the infant's stool and tongue swab,

correct?

Response:

Yes, this is correct. Microbes that were shared between infant and mother stool samples were at higher relative abundance on average than those microbes that were not shared, at least until the infant reached 12 months of age. Similarly, microbes that were shared between the infant and mother's tongue swabs were at higher relative abundance on average than those microbes that were not shared until the infant reached 12 months of age. The main text has been updated to make this point more clear.

→ Line 214: The authors state the ASVs in the infant stool fit the neutral model moderately well. But the value is closer to 0 than 1. Wouldn't one have expected the R2 value to be closer to 1 or at least above 0.5 to say that it fit the neutral model moderately well?

Response:

We have replaced R² values with the RMSE (root mean squared error) to give a more interpretable and objective description of how well the observations fit the model's predictions. We have also removed subjective statements about how well data fit the model, and instead simply report the values and let the reader decide.

→ Line 216: I would be interested in which taxa experienced positive selection. These would likely be "beneficial" in some way, and it might be useful to know which ones were included in this group.

Response:

The taxa on which there was consistent evidence of selection (both positive and negative) are included in the heatmap in Fig. 3c.

→ Line 219: When the mother pool was used, it was a poorer model fit (R2 = -0.29). The model fit for mother tongue was 0.19 and the authors have indicated that this still fit the neutral model. Is an R2 above 0.1 considered a relatively good fit that indicates neutral dispersal; whereas, values that are very close to 0 or negative, indicative of either positive or negative selection? Additionally, in Line 229, I would have thought that an R2 of 0.19 might also be indicative of selection. Perhaps I am just misunderstanding this.

Just curious if these analyses were separated out by groups of infant age, I wonder how the R2 values might change?

Response:

As discussed above, we have removed the analysis in which we use the mothers' samples as a source pool. More generally, to avoid erroneous interpretations of R², we have replaced it with RMSE as a way of quantifying goodness-of-fit and have removed value-laden judgments from the text. In the

context of a measurement of goodness-of-fit and not variation explained, the interpretation of either metric is largely qualitative and better interpreted in the context of how it differs between groups.

→ Line 228: From my viewpoint, I wouldn't say that the model still fit "relatively well" for the mother's tongue swab pool.

Response:

We have revised the text of our manuscript to be more objective in our reporting of the results of model fitting, and now refrain from making subjective judgments about what constitutes a "good" or "poor" model fit. In addition, we now report RMSE, which similarly measures goodness-of-fit, while hopefully avoiding confusion for readers who may assume that R^2 is a measurement of variation explained.

→ Line 286+: Although it's interesting to speculate that market access is a driving force behind variation in microbial community structures, there is a plethora of other differences among these villages. I would be extremely careful here to suggest these differences are due to market access or market integration. Can you further refine these analyses to take into account some of the other differences between these communities?

Response:

We agree that caution is warranted in drawing conclusions around this issue. The fact that adults from different river village types (proximal versus distal) have distinct microbial membership (Fig. 5c), suggests that market access may be a primary driver of those differences. That said, the current sample set with river and forest villages does not allow us to distinguish between the effects of season and geography from other potential differences among Tsimane villages.

→ Line 293-295: Which sites have the higher intake and higher levels? I am guessing that these are referring to the proximal river sites?

Response:

Yes, the proximal river villages have higher intake of processed foods and higher levels of urinary phthalates than the distal villages. The main text has been revised to make these points more clear.

→ Line 300-302: Might season influence the outcome of differences between village ecotypes? Were samples treated the same from collection through sample processing? Providing a statement in this regard would be helpful to understand the potential factors

that might be at play in influencing the microbial composition.

Response:

We have added text to the discussion section to address the unavoidable confounding of season with other factors:

“Microbial community differences across villages may also reflect local ecological influences, including water sources, pathogen exposures, and dietary composition of foraged foods. However, logistical constraints in our study design necessitated that river and forest villages be sampled during different seasons (rainy versus dry), which confounds seasonal versus geographical influences on diet and pathogen exposures.”

In addition, we have added statements to indicate that all samples were treated identically from collection through sample processing.

“Field specimen collection protocols were devised with the logistical challenges of this population in mind, and were consistent across the 2012-2013 and 2009 cohorts.”

“All samples from both cohorts were shipped on dry ice to a laboratory in the United States, where they remained at -80°C until they were processed.”

→ Line 364-367: What is the “same body site” that is being referred to in this sentence? Not sure the ref supports the statement that “other maternal body sites is relatively rare.” The Ferritti paper describes contributions from maternal vaginal, skin, oral, and gut, with maternal vaginal and skin contribution decreasing during the first few days postpartum.

Response:

We agree that “relatively rare” is an overstatement, although Ferretti *et al.* (2018, *Cell Host and Microbe*) do demonstrate that the majority of microbial strains observed in the infant’s stool that are also found in their mother are found in their mother’s stool, and that the majority of microbial strains observed in the tongue swabs from the infant that are also in their mother are in their mother’s tongue swabs. We have updated the main text of the manuscript to describe their findings more succinctly.

→ Line 367-370: Understanding the contribution of ICABs on infant microbial assembly, structure, and dynamics is very intriguing. Are there any data that can be presented to support that levels of various ICABs are related to microbial composition?

Response:

We interpret this question as asking if variance in the amount or intensity of an ICAB within the Tsimane population can be related to infant microbial composition. Unfortunately, we cannot test this directly using the current

dataset. The ICABs we describe in the introduction (birth mode, breastfeeding, co-sleeping, maternal contact, food premastication) are either constant or vary only minimally among the Tsimane. Fine scale behavioral data (at the level of the individual through time) were not collected. However, the analysis presented in Figure 4 is our attempt at comparing the Tsimane style of maternal caregiving with those from other cultures.

Our results should be interpreted as evidence of maternal-infant microbial dynamics when these types of ICABs are conserved. These results may be used as a baseline for interpreting variation in other populations. That is, future research might predict that in populations with more variance in these ICABs, specific dyads with ICAB profiles that are more similar to those observed in this study (for example, Tsimane mothers that have moved to larger cities but still practice some traditional ICABs) should exhibit more similar microbial dynamics than dyads at the opposite end of the ICAB spectrum. Further study in populations with more ICAB variance will be needed to tease apart the effect of ICABs on microbiota assembly.

Materials and Methods

→ Line 412-413: Please provide your UCSB IRB protocol #s.

Response:

The IRB protocol number has been added to the main text of the manuscript.

→ Line 423: Please provide a further explanation of what mixed longitudinal design means.

Response:

In a traditional or pure longitudinal sampling design, all research subjects are sampled on multiple, consistent times over a defined period. In a cross-sectional study, subjects are studied at one specific point in time. For the current study, enrolled children were first sampled in a cross-sectional manner, but in addition, a subset was re-sampled up to 8 times during the field season (see Figure 1a for details). This sample collection approach was implemented due to the fact that investigators were only able to collect samples during their field season. We used the term, “mixed longitudinal design” to describe this pattern.

→ Was the longitudinal sampling spread across all 6 sites or for example, were the samples collected during early months after birth collected from one site and later after birth collected from another site?

Response:

Investigators enrolled all children under the age of two, and sampled them when they were available during the 2012 – 2013 field season. No attempt was made to normalize for infant age across sites, because in most cases, every infant living in the village was enrolled. Detailed information on the age of each subject at the time of sampling, as well as the village site at which they resided, is now available in Supplementary Table 2.

→ Line 433+: Please provide additional information related to the sample collection containers, including whether they were sterile. What instructions were the subjects given in terms of how to collect their samples? Were samples collected from the ground? Were all samples treated the same? Also, additional information is needed as to how the fecal samples were transferred into the containers and whether there was any sort of cleansing step prior to this.

Response:

The main text of the manuscript has been expanded to provide further details on the sample collection process:

“Field specimen collection protocols were devised with the logistical challenges of this population in mind, and were consistent across both the 2012-2013 and 2009 cohorts. Tsimane families do not have plumbing, do not use pit toilets or diapers, and do not have access to refrigeration. Fecal samples were collected in sterile urine specimen cups. Cups were given to mothers the day before sample collection, in separate plastic bags, with specimen cup lids identified by different symbols for mother and infant. Mothers were instructed to fill the collection cups with the first bowel movement of the following day, and to keep collection cups out of the sun. Bags included sheets of paper onto which to defecate and small plastic spoons with which to handle feces, though we could not be sure this protocol was followed. Mothers collected infant feces directly or from swaddling clothes.

Researchers returned to participants’ homes between 7-9 am to collect the specimens, making 1-2 return visits as necessary until approximately midday. Samples were transported to a field laboratory in coolers with reusable ice packs within 1-2 hours of collection. Samples were homogenized in the collection cup and then partitioned out into 2ml sterile cryotubes using nonsterile wooden tongue depressors. Cryotubes were immediately stored in liquid nitrogen. Oral samples were collected from participants by research staff with a buccal cell collection swab (Catch-All™ Sample Collection Swab; Epicentre^R), transferred to cryotubes and stored in liquid nitrogen. All samples were shipped on dry ice to a laboratory in the United States, where they remained at -80°C until they were processed.”

→ Line 440: Please provide additional information regarding what beads were used.

Response:

The beads were 0.7 mm garnet beads that were supplied in the MOBIO PowerSoil DNA extraction kit by the manufacturer. The main text of the manuscript has been updated to indicate this.

→ Line 443: What is an error-correcting barcode?

Response:

The set of 8 nucleotide error-correcting barcodes that we used here are all sufficiently different from each other that a single sequencing error can be detected and corrected, while two sequencing errors can be detected but not corrected. Details can be found in the referenced article (Hamady *et al.*, 2008, *Nature Methods*), which was also added to the main text.

→ Line 451:

Current study – DNA extracted using MoBio PowerSoil-htp kit with bead-beating; V4 region using 515F and 806R

Bangladesh (Subramanian *et al.* 2014) – DNA isolated by beadbeating in phenol/chloroform, purified with QIAquick column. Not totally clear what positions the V4 primers were.

Finland (Vatanen *et al.* 2016) – DNA isolated using the QIAamp DNA Stool Mini Kit. PCR following Caporaso *et al.* 2012)

England – (Goodrich *et al.* 2016) – unclear how DNA isolated; references Goodrich *et al.* 2014 but no details here. Amplified V4 using primers 515F and 806R); sequenced on MiSeq 2x250 bp paired-end sequencing

Although all 3 studies sequenced the V4 region, there were differences in the DNA extraction procedure. It is fairly well documented that different DNA extraction methods can yield different results; thus, although the authors state that methods were relatively similar, caution and/or mention of this as a limitation to these comparisons should be mentioned.

Response:

We concur with the reviewer that variation in the extraction method can sometimes make comparisons between studies problematic. However, our emphasis was on the striking similarities between the studies, especially among the very early life stool samples, despite those methodological differences.

→ Were there any negative controls done with the extraction and PCR and if so, were they sequenced? If so, did the authors consider running decontam to identify potential contaminants?

Response:

Yes, there were negative controls run and sequenced along with these samples. However, they typically yielded so few reads (<100) that they were considered not reliable and were therefore excluded from any downstream analyses.

→ Were any positive controls (e.g. mock communities?) extracted and/or was the DNA amplified with PCR? If so, were they sequenced?

Response:

No mock communities or positive controls were extracted or sequenced in this investigation.

→ Line 455: Why were Archaea included?

Response:

While it was not our explicit goal to profile the archaeal constituents of the microbiome, the 16S rRNA primer pair that we used (515F/806R) is well known for amplifying some archaeal sequences, including those from the prevalent gut commensal, *Methanobrevibacter smithii*. Indeed, including archaea in this analysis was key to the observations described in Figure 6.

→ Line 471: This is the first mention of the MDD food groups. There is a nice description of this FFQ in the results section, but I would have expected it to be included in the methods section (most people still read the methods before they read the results).

Response:

We have moved the detailed description of the minimum dietary diversity score to the Methods section, and added a parenthetical note in Results directing the reader to Methods if they seek additional details.

→ Line 496: Similarly, I would have appreciated a more extensive discussion of the “regional species pool” in the methods section rather than the results section.

Response:

Explanation of the regional species pool has been expanded.

FIGURES

→ Figure 1 Not sure what the y axis is on Figure 1a.

Response:

The y-axis has now been labeled “Dyad” to indicate that each horizontal line represents a mother-infant dyad.

→ Text mentions statistical outcomes for Figure 1e, and it would be nice if they were also on the figure.

Response:

Only the components of the MDD that are statistically significant are displayed in Figure 1e. We have added asterisks to denote the level of significance for each dietary factor, * denotes $p < 0.05$, ** denotes $p < 0.01$.

→ Figure 1e: What do the circles represent? Are these ASVs or are they infant stool taxa composition of various samples?

Response:

The circles in Figure 1e (as well as in Figs. 1a,b,d) represent infant stool samples, colored by the age of the infant (darker = older).

→ Figure 2: In order to be counted as shared, was the ASV shared between at least one individual mother/infant dyad or multiple mother/infant dyads?
Or shared between any mother sample and infant sample?
And were they compared across all samples from the mother / infant dyad? Or just the matched timepoint?

Response:

In Fig. 2a and 2c, each individual sample is represented by both a blue and red data point. The ASVs in each stool sample or tongue swab were binned as either “Shared” -- those that were found in any of their own mother’s samples at the same body site), or as “Not Shared”. Since stool samples from dyads were not always collected on the same day, we could not use matched time points.

→ Figure 3 I would recommend splitting this figure into at least 2 separate figures. It’s next to impossible to read the ASV/taxa designations.

Response:

Figure 3 has now been split into a revised Fig. 3 and Supplemental Fig. 7.

→ Figure 3a: Would it be possible to make the color for the “Below prediction” closer to the row color in 3b? This might help in relating the two together.

Response:

The colors now used to indicate below and above prediction groups are the same in panels 3b (related to the old Fig. 3a) and 3c (similar to the old Fig. 3b).

→ Figure 3b. I wonder since the authors are specifying specific ASVs, could they

highlight these somehow either with an arrow or a square around the circle?

Response:

If we are interpreting this comment correctly, the reviewer is suggesting that we identify the data points in the scatter plot (updated Fig. 3b, previous Fig. 3a) that link them to the heatmap (updated Fig. 3c, previous Fig. 3b). We believe that the updated Figure 3 now does a better job in relating the groups of ASVs that were consistently under positive or negative selection (yellow and green, respectively) to their model fit. Of note, the data displayed in the updated Fig. 3b are limited to the one model of 1,000 replicates that had the smallest RMSE, whereas the updated Fig. 3c displays only ASVs that were consistently under positive or negative selection in $\geq 90\%$ of bootstrapped replicates.

→ Figure 4a: Would help to color the Finnish infants and English adults a different color. And maybe even include a different shape?

Response:

In response to reviewer comments, the English adults have been removed from the analysis.

→ Figure 4a: What do the lines linking the samples represent?

Response:

The lines in the original version of this Figure linked each sample with the principal curve, but the latter has now been removed.

→ Figure 4e: Once again, font is small.

Response:

The font size has been increased.

→ Figure 5 Can the authors make the symbols in Figure 5a the same as those in Figure 5b?

Response:

The circles in Fig. 5a were scaled to represent the number of samples collected at each village, so if we were to make them different shapes then direct size comparisons would be less visually intuitive. However, we agree that consistency across panels is important for clarity, and have decided to change the shapes in the map in Fig. 5a to match subsequent figures. In addition, the number of samples collected per village is also available in Table 1, so the intended information has not been lost.

→ Figure 5d: Font of ASV numbers are very small.

Response:

The overall size of Fig. 5d (now Fig. 5c) has now been increased and the layout improved, so that ASV numbers are larger and more readable.

→ Figure 6a: What does the length of the edges between nodes represent?

Response:

Additional details regarding construction and display of the network have been added to Methods. In short, node placement was determined by a common iterative force-directed algorithm that seeks to minimize the energy (i.e., edge length) of the global network layout. More information can be found in Fruchterman, T. M. J., & Reingold, E. M. (1991). Graph Drawing by Force-Directed Placement. *Software: Practice and Experience*, 21(11).

The manuscript text now reads:

“Network analysis was performed using the SPIEC-EASI function in the `SpiecEasi` R package (version 1.0.6) to infer an underlying graphical network model using both sparse neighborhood and inverse covariance selection⁵¹. ASVs were removed that were not observed at least 10 times in at least 20% of Tsimane adult stool samples. The network was constructed with the SPIEC-EASI network inference algorithm using the Meinshausen-Buhlmann Neighborhood Selection method. The sizes of the nodes were scaled to represent their degree of connectedness. The distance between the nodes was determined using the force-directed layout algorithm by Fruchterman and Reingold as implemented in the `layout.fruchterman.reingold` function, while community sub-structure was determined using the multi-level modularity optimization algorithm for finding community structure as implemented in the `cluster_louvain` function, both available in the `igraph` R package (version 1.2.4.1).”

→ Supplementary Figure 3c: Missing ‘Left panel – infant stools, right panel – infant tongue swabs’

Response:

Thank you. This has been added to the figure legend for Supplementary Figure 3c.

→ Supplementary Figure 5: The shape of the curve is different among the different populations. What, if any, is the significance of this?

Response:

The shape of the curve (i.e., the predictions made by the model) will vary depending on the dataset with which it is built.

Reviewer #4 (Remarks to the Author):

This manuscript reports the results of a longitudinal study of the oral and gut microbiomes of 52 infant-mother pairs in an indigenous Brazilian population. Lifestyles of the Tsimane people is undergoing change including increased access to market good and medicine, factors that have direct potential to influence their microbiomes. The manuscript is written well and will be of most interest to microbiome researchers, but is accessible to a broader scientific audience.

The authors focus on infant care associated behaviors that are likely to increase maternal to infant transmission. In particular, pre-masticating foods and preparation of chicha – a fermented drink inoculated with saliva – have the obvious potential to influence colonization of the infant gut. Results are also compared with microbiota from infants in Bangladesh and Finland. The conclusion that changes in lifestyle are affecting the trajectory of the gut microbiome is intriguing and of broad interest. Many of the other findings confirm published results and extend them to the Tsimane population.

The major concern I have is in the interpretation and presentation of the neutral model. While application of the neutral model is appropriate for this study, the interpretation of results from the neutral model, namely that "...the majority of microbes colonizing infants from different countries were neutrally distributed" could easily be misinterpreted by readers. The neutral model makes the provocative assertion that all individuals, in this case amplicon sequence variants (ASVs) are ecologically identical and that their occurrence in the microbiome is based solely on the pool of ASVs available to colonize the gut or mouth. A neutral distribution in this case does not mean that there is no selection, but rather that selection in the larger pool is the same as selection in the target population. The gut is a strongly selective environment and that may be missed in the current presentation.

Response:

This is a very important point, and we thank the reviewer for helping to clarify some of the issues at hand. In order to improve clarity, we have added Fig. 3a and Supplementary Figure 7a, as well as added text to the first paragraph of the "Microbiota Assembly Rules" section:

"The NCM predicts the prevalence of each microbe given its average relative abundance in a regional species pool (RSP). Microbes that fit the prediction are inferred to have assembled neutrally from the RSP, while microbes at higher or lower prevalence are inferred to have been under local positive or negative ecological selection, respectively. However, species whose

distributions fit the neutral model may still have experienced selection, if the selection was of similar direction and magnitude to the selection present in the RSP. In addition, the NCM assumes that all microbes in a regional species pool have an equal per capita ability to disperse to a local area, and once established, to have equal per capita fitness, growth, and death rates³⁴.”

REVIEWERS' COMMENTS:

Reviewer #1 (Remarks to the Author):

I thank the authors for addressing my questions. I have no further comments at this stage.

Reviewer #2 (Remarks to the Author):

Sprockett et al. have adequately addressed all of my major concerns regarding the manuscript. In particular I think that the new simpler comparisons of the overlaps between mother and infant pairs give an important insight into the dataset. I found that the revised neutral modelling approach and its description are now more easily interpreted but, as mentioned in my previous review, I have no experience applying these approaches myself. I would also like to acknowledge the excellent effort to share reproducible code associated with the paper, I was very impressed by the markdown files available on the Stanford repository. Overall, I recommend the revised manuscript for publication and just have a few additional suggestions regarding some of new analyses.

1. The reduced international comparisons improve the manuscript and I appreciate the addition of the taxonomy-based distance measures in Supplementary Figure 8. However, on line 918 it states that the 3 microbiota trajectories diverged with age in the different countries. In the new UniFrac plots this pattern is less apparent, particularly between the Bolivian and Bangladeshi populations in the Unweighted UniFrac and between all 3 in the Weighted UniFrac comparison. I don't think this contradicts the original finding but it might be interesting to discuss this disparity in the manuscript. This might indicate that population-level differences are more pronounced at the species/strain level but more consistent based on high-level taxonomic functions, which the UniFrac measures would suggest. This is relevant to the paper's discussion of the sources of taxa in an individual sample.

2. It would be nice to add a sentence regarding your response to my previous comment that the Christensenellaceae subnetwork was not associated with BMI in the Tismaine cohort to the manuscript (around line 1195), as it is pitched as an investigation of obesity in this group.

Reviewer #3 (Remarks to the Author):

I'd like to thank the authors for their thoughtful and thorough responses to reviewers' comments. Overall the manuscript is much improved. I have a few remaining general comments.

I remain a bit concerned about the emphasis that you place on market integration and culture change as the driver of these differences. Your abstract and conclusion, although you did add an additional statement, still highlights this as your finding. I understand that you didn't design the study to test for geographic or seasonal differences, but I feel you should soften your conclusions and provide a more thorough treatment of other possible explanations. Without ruling out the other factors, I think you should focus on the differences across the localities rather than making the untested link to cultural change a primary conclusion.

Additionally, the description of your sample is still a bit confusing. The first 4 comments below are questions about sample description in text and in your figures. I listed these but this comment

apply across the manuscript and the supplementary materials. A chart or an additional column in your sample description table that walks a reader through each analysis and how many individuals/samples were included may help. It remained challenging to determine as I read the manuscript, particularly when I saw that you reduced the pairs to 47 but the sample #s weren't updated. I may be misunderstanding, but if you can make it easier on the reader that would help.

Supplementary Figure 1. It looks like there are more infants than adults. Does this just include the matched pairs? Can you provide a sample description in the caption?

Supplementary Figure 2. Please add sample description.

Line 92-93 - Please confirm that the sample ns didn't change when you lowered the number of mother infant pairs.

Line 443 – 446. In these lines you present the samples that you have, not what you analyzed. You should present both.

Line 475 – When you say directly, do you mean from the skin? If so, I'd add directly "from the skin" or swaddling clothes.

Reviewer #4 (Remarks to the Author):

[No further comments for authors]

Sprockett *et al.*, “Microbiota Assembly, Structure, and Dynamics Among Tsimane Horticulturalists of the Bolivian Amazon”

Responses to Reviewers

Reviewer #1: No responses needed.

Reviewer #2:

1. The reduced international comparisons improve the manuscript and I appreciate the addition of the taxonomy-based distance measures in Supplementary Figure 8. However, on line 918 it states that the 3 microbiota trajectories diverged with age in the different countries. In the new UniFrac plots this pattern is less apparent, particularly between the Bolivian and Bangladeshi populations in the Unweighted UniFrac and between all 3 in the Weighted UniFrac comparison. I don't think this contradicts the original finding but it might be interesting to discuss this disparity in the manuscript. This might indicate that population-level differences are more pronounced at the species/strain level but more consistent based on high-level taxonomic functions, which the UniFrac measures would suggest. This is relevant to the paper's discussion of the sources of taxa in an individual sample.

Response:

We thank the reviewer for this useful feedback. We have added text highlighting the differences between the microbiota maturational trajectories using the different distance metrics.

2. It would be nice to add a sentence regarding your response to my previous comment that the Christensenellaceae subnetwork was not associated with BMI in the Tsimane cohort to the manuscript (around line 1195), as it is pitched as an investigation of obesity in this group.

Response:

We have added a statement indicating that we did not observe an association between the relative abundance of these taxa and adult BMI. We also added a citation to a recent manuscript demonstrating that Christensenellaceae and *Methanobrevibacter* participate in H₂ syntrophy.

Reviewer #3 (Remarks to the Author):

I'd like to thank the authors for their thoughtful and thorough responses to reviewers' comments. Overall the manuscript is much improved. I have a few remaining general comments.

I remain a bit concerned about the emphasis that you place on market integration and culture change as the driver of these differences. Your abstract and conclusion, although you did add an additional statement, still highlights this as your finding. I understand that you didn't design the study to test for geographic or seasonal differences, but I feel you should soften your conclusions and provide a more thorough treatment of other possible explanations. Without ruling out the other factors, I think you should focus on the differences across the localities rather than making the untested link to cultural change a primary conclusion.

Response:

We thank the reviewer for this very helpful feedback. Throughout the manuscript, we have softened or removed statements that directly relate differences in microbiome composition to market integration or market access, and have further emphasized that differences in season and/or local ecology may also contribute to those differences. Furthermore, we have more specifically quantified differences in market access between village types, so that the reader may be more aware of their magnitude.

Additionally, the description of your sample is still a bit confusing. The first 4 comments below are questions about sample description in text and in your figures. I listed these but this comment apply across the manuscript and the supplementary materials. A chart or an additional column in your sample description table that walks a reader through each analysis and how many individuals/samples were included may help. It remained challenging to determine as I read the manuscript, particularly when I saw that you reduced the pairs to 47 but the sample #s weren't updated. I may be misunderstanding, but if you can make it easier on the reader that would help.

Response: We have tried our best to clarify and simplify further the sample description.

Supplementary Figure 1. It looks like there are more infants than adults. Does this just include the matched pairs? Can you provide a sample description in the caption?

Response:

Supplementary Figure 1 includes samples from the 47 dyads only. However, since many of the infants were sampled more frequently than the adults (see Figure 1a for details), there are more infant samples than adult samples displayed here. We have added a sample description to the figure legend for clarification.

Supplementary Figure 2. Please add sample description.

Response:

We have added a sample description to this figure legend for clarification.

Line 92-93 - Please confirm that the sample ns didn't change when you lowered the number of mother infant pairs.

Response:

Thank you. These numbers have now been updated to reflect the numbers of samples that were included in the final analysis.

Line 443 – 446. In these lines you present the samples that you have, not what you analyzed. You should present both.

Response:

The number of samples analyzed has been added.

Line 475 – When you say directly, do you mean from the skin? If so, I'd add directly "from the skin" or swaddling clothes.

Response:

"from the skin" has been added to the text.

Reviewer #4: No responses needed.